# Non-Euclidean Gradient Descent Operates at the Edge of Stability

## Abstract

The Edge of Stability (EoS) is a phenomenon where the sharpness (largest eigen-value) of the Hessian converges to $2/\eta$ during training with gradient descent (GD) with a step-size $\eta$. Despite violating classical smoothness assumptions, EoS has been widely observed in deep learning, but its theoretical foundations remain in-complete. We propose a framework for analyzing EoS of non-Euclidean GD using directional smoothness (Mishkin et al., 2024), which naturally extends to non-Euclidean norms. This approach allows us to characterize EoS beyond the stan-dard Euclidean setting, encompassing methods such as $\ell_\infty$-descent, Block CD, Spectral GD, and Muon without momentum. We derive the appropriate measure of the generalized sharpness under an arbitrary norm. Our generalized sharpness measure includes previously studied vanilla GD and preconditioned GD as special cases. Through analytical results and experiments on neural networks, we show that non-Euclidean GD also exhibits progressive sharpening followed by oscilla-tions around the threshold $2/\eta$. Practically, our framework provides a single, geometry-aware spectral measure that works across optimizers, bridging the gap between empirical observations and deep learning theory.

## 1 Introduction

In supervised settings, training machine learning models is posed as empirical risk minimization $\min_{\mathbf{w}\in\mathbb{R}^d}\mathcal{L}(\mathbf{w})$, where $\mathbf{w}\in\mathbb{R}^d$ are the neural network's parameters, and $\mathcal{L}(\mathbf{w})$ is the full-batch loss, which we assume is bounded below by $\mathcal{L}^* > -\infty$. In deep learning, $\mathcal{L}$ is typically nonconvex and highly structured (Li et al., 2018; Kim et al., 2024). Nevertheless, first-order methods such as SGD and its adaptive variants (Duchi et al., 2011; Kingma & Ba, 2014) are the workhorses of practice and scale effectively to large models, despite a limited theoretical understanding of their success.

Full-batch gradient descent (GD) serves as the canonical proxy for analyzing gradient-based training. Classical results for $L$-smooth convex objectives guarantee descent for step sizes up to $2/L$. In contrast, recent empirical work reveals a characteristic two-phase behavior when deep networks are trained with GD. In the initial phase, called the progressive sharpening phase, the loss $\mathcal{L}(\mathbf{w}_t)$ decreases monotonically while the sharpness $S(\mathbf{w}_t) := \lambda_{\max}(\nabla^2\mathcal{L}(\mathbf{w}_t))$ grows. This is followed by the edge-of-stability (EoS) phase, where the loss behaves non-monotonically yet decreases over longer horizons, while the sharpness hovers near the threshold $2/\eta$ (Cohen et al., 2021).

The EoS phenomenon has been found to extend beyond vanilla GD. Cohen et al. (2022) showed that adaptive preconditioning methods such as Adagrad and Adam exhibit an EoS characterization that revolves around the top eigenvalue of the *preconditioned* Hessian, while Long & Bartlett (2024) showed that SAM obeys a certain EoS characterization as well. Despite these advances, the question of how EoS generalizes to other optimizers remains underexplored. In this work, we investigate how the EoS phenomenon carries over to a broad family of optimization algorithms: that of non-Euclidean gradient descent with respect to an arbitrary norm.

**Definition 1.1.** For a norm $\|\cdot\|$ and a step-size $\eta > 0$, the associated non-Euclidean GD method is given by the minimization of the regularized linearization around the current point $\mathbf{w}_t$:

$$\mathbf{w}_{t+1} = \operatorname*{argmin}_{\mathbf{y}} \mathcal{L}(\mathbf{w}_t) + \langle \nabla\mathcal{L}(\mathbf{w}_t), \mathbf{y} - \mathbf{w}_t \rangle + \tfrac{1}{2\eta}\|\mathbf{y} - \mathbf{w}_t\|^2$$

$$= \mathbf{w}_t - \eta \|\nabla \mathcal{L}(\mathbf{w}_t)\|_* (\nabla \mathcal{L}(\mathbf{w}_t))_*, \tag{1}$$

where the *dual norm* $\|\nabla \mathcal{L}(\mathbf{w}_t)\|_*$ and *dual gradient* $(\nabla \mathcal{L}(\mathbf{w}_t))_*$ are defined as:

$$\|\nabla \mathcal{L}(\mathbf{w}_t)\|_* := \max_{\|\mathbf{y}\|=1} \langle \nabla \mathcal{L}(\mathbf{w}_t), \mathbf{y} \rangle, \quad (\nabla \mathcal{L}(\mathbf{w}_t))_* := \operatorname*{argmax}_{\|\mathbf{y}\|=1} \langle \nabla \mathcal{L}(\mathbf{w}_t), \mathbf{y} \rangle. \tag{2}$$

We let $\mathbf{d}_t := \|\nabla \mathcal{L}(\mathbf{w}_t)\|_* (\nabla \mathcal{L}(\mathbf{w}_t))_*$ denote the update "direction" (i.e. the update without $\eta$).

This formulation reduces to vanilla GD when the norm $\|\cdot\|$ is taken to be the $\ell_2$ norm. It also subsumes methods not previously studied by prior work on EoS such as $\ell_\infty$-descent (for $\|\cdot\| = \ell_\infty$) and Spectral GD (for $\|\cdot\| = \|\cdot\|_{2\to 2}$) (Carlson et al., 2015) (which underlies the popular Muon method (Jordan et al., 2024)), as well as Block CD (Nesterov, 2012) and other coordinate descent variants.

Sometimes, the dual norm is omitted from the update (1). We refer to the resulting algorithm as *normalized* non-Euclidean GD:[1]

**Definition 1.2.** For a norm $\|\cdot\|$ (not necessarily the $\ell_2$ norm) and a step-size $\eta > 0$, the associated *normalized* non-Euclidean GD method is given by

$$\mathbf{w}_{t+1} = \mathbf{w}_t - \eta (\nabla \mathcal{L}(\mathbf{w}_t))_*, \tag{3}$$

where the dual gradient $(\nabla \mathcal{L}(\mathbf{w}_t))_*$ is defined in (2).

When $\|\cdot\|$ is the $\ell_\infty$ norm, this formulation recovers SignGD (Bernstein et al., 2018), and when $\|\cdot\|$ is the spectral norm $\|\cdot\|_{2\to 2}$, it recovers Muon (Jordan et al., 2024). Our main contributions are summarized as follows:

1. We identify that an intermediary quantity called directional smoothness $D^{\|\cdot\|}(\mathbf{y}, \mathbf{w})$ (Mishkin et al., 2024) can be used to study the dynamics of sharpness and the EoS. Directional smoothness is an average curvature between two consecutive iterates.

2. Through a simple identity, we show that if the loss decreases, and the gradient norm squared is approximately stable, then directional smoothness *must* increase up to $2/\eta$. Sharpness is an (approximate) upper-bound on directional smoothness, thus when directional smoothness increases up to $2/\eta$, so will sharpness. Furthermore, if the loss oscillates, then directional smoothness must also oscillate around $2/\eta$.

3. Extending directional smoothness beyond Euclidean norm, we define a generalized sharpness $S^{\|\cdot\|}$ of GD under any norm $\|\cdot\|$. In the special cases of Euclidean and preconditioned GD, this measure recovers previously established notions of sharpness.

4. Across MLPs, CNNs, and Transformers architectures, we observe that $S^{\|\cdot\|}$ hovers around the stability threshold $2/\eta$, demonstrating EoS behavior in diverse architectures.

5. To shed light on the mechanism underlying this behavior, we analyze the dynamics of non-Euclidean GD on quadratic objectives.

## 1.1 RELATED WORKS

The EoS phenomenon was first documented for vanilla GD with step-size $\eta$, where the sharpness (the maximum Hessian eigenvalue) was observed to hover near the stability threshold $2/\eta$ (Cohen et al., 2021). This initial work also extended empirical observations to GD with momentum and provided intuition for EoS on quadratic objectives. Building on this, Arora et al. (2022) gave a mathematical analysis of the implicit regularization that arises at EoS, showing that in non-smooth loss landscapes the updates of normalized GD follow a deterministic flow constrained to the manifold of minimal loss. A subsequent study by Song & Yun (2023) demonstrated empirically that GD trajectories align with a universal bifurcation diagram during EoS, while Damian et al. (2022) identified self-stabilization as the key mechanism: a cubic term in the Taylor expansion along the top Hessian eigenvector introduces negative feedback that drives sharpness back toward $2/\eta$ whenever it exceeds the threshold. Beyond the stability plateau, Ghosh et al. (2025) analyzed loss oscillations in deep linear networks,

---

[1]We refer to algorithms that satisfy Def. 1.1 and 1.2 for $\ell_\infty$ norm as $\ell_\infty$-descent and SignGD respectively.

demonstrating that they happen in a low-dimensional subspace whose dimension depends on the step-size $\eta$. Finally, several works connect EoS with the catapult mechanism observed in training with a large learning rate (Lewkowycz et al., 2020; Zhu et al., 2024; Kalra & Barkeshli, 2023).

The phenomenon has also been studied for preconditioned and adaptive methods. Cohen et al. (2022) showed that the sharpness of the preconditioned Hessian stabilizes at the same threshold for methods such as AdaGrad and RMSprop. Meanwhile, Long & Bartlett (2024) conducted a stability analysis of SAM (Foret et al., 2020) on quadratics, empirically showing that SAM operates at the edge of stability. Extensions beyond full-batch GD include Lee & Jang (2023), who analyzed the interaction between batch-gradient distributions and loss geometry to extend EoS to SGD, and Andreyev & Beneventano (2024), who proposed an alternative stochastic counterpart of EoS.

Despite this progress, most prior studies have focused on a narrow family of algorithms (e.g., vanilla GD, preconditioned GD, or SAM), leaving a fundamental gap in our understanding of spectral properties and raising the question of whether these insights extend to substantially different optimization methods such as Muon (Jordan et al., 2024) and SignGD (Bernstein et al., 2018). In this work, we close this gap by introducing a unified framework for analyzing EoS across optimization algorithms, leveraging the recent insight that many methods can be interpreted as variants of steepest descent under an appropriate norm (Bernstein & Newhouse, 2024).

## 2 PROGRESSIVE SHARPENING AND DIRECTIONAL SMOOTHNESS

Classical descent guarantees for GD rely on global $L$-smoothness, but such bounds are often too pessimistic for neural networks (Zhang et al., 2019). Instead, we adopt a local, trajectory-aware notion of directional smoothness (Mishkin et al., 2024).

**Definition 2.1.** We call a function $D^{\|\cdot\|}(\mathbf{w}_t, \mathbf{w}_{t+1})$ a valid *directional smoothness* at iteration $t$ if

$$\mathcal{L}(\mathbf{w}_{t+1}) \leq \mathcal{L}(\mathbf{w}_t) + \langle \nabla \mathcal{L}(\mathbf{w}_t), \mathbf{w}_{t+1} - \mathbf{w}_t \rangle + \frac{D^{\|\cdot\|}(\mathbf{w}_t, \mathbf{w}_{t+1})}{2} \|\mathbf{w}_{t+1} - \mathbf{w}_t\|^2, \tag{4}$$

where $D^{\|\cdot\|}(\mathbf{w}_t, \mathbf{w}_{t+1})$ depends only on the behavior of the loss $\mathcal{L}$ along the chord $[\mathbf{w}_t, \mathbf{w}_{t+1}]$.

Mishkin et al. (2024) provide several examples of the directional smoothness. In this work, we choose the tightest one

$$D^{\|\cdot\|}(\mathbf{w}, \mathbf{y}) := \frac{\mathcal{L}(\mathbf{y}) - \mathcal{L}(\mathbf{w}) - \langle \nabla \mathcal{L}(\mathbf{w}), \mathbf{y} - \mathbf{w} \rangle}{\frac{1}{2}\|\mathbf{y} - \mathbf{w}\|^2}, \tag{5}$$

which makes (4) hold with equality. Although this quantity might not be positive (and thus falls outside the positivity requirements of Mishkin et al. (2024)), positivity is not required in the following presentation. Substituting one step of non-Euclidean GD into (4) yields

$$\mathcal{L}(\mathbf{w}_{t+1}) = \mathcal{L}(\mathbf{w}_t) - \eta \langle \nabla \mathcal{L}(\mathbf{w}_t), \mathbf{d}_t \rangle + \frac{\eta^2}{2} D^{\|\cdot\|}(\mathbf{w}_t, \mathbf{w}_{t+1}) \|\mathbf{d}_t\|_*^2$$

$$= \mathcal{L}(\mathbf{w}_t) - \eta \left(1 - \frac{\eta}{2} D^{\|\cdot\|}(\mathbf{w}_t, \mathbf{w}_{t+1})\right) \|\nabla \mathcal{L}(\mathbf{w}_t)\|_*^2. \tag{6}$$

Whenever $\|\nabla \mathcal{L}(\mathbf{w}_t)\|_* > 0$, the loss decreases if *and only if*

$$\mathcal{L}(\mathbf{w}_{t+1}) \leq \mathcal{L}(\mathbf{w}_t) \iff D^{\|\cdot\|}(\mathbf{w}_t, \mathbf{w}_{t+1}) \leq \frac{2}{\eta}. \tag{7}$$

The equivalence in (7) justifies the progressive sharpening of the directional smoothness. Note that in deep learning experiments where EoS is observed, the gradient norm remains non-zero (Defazio et al., 2023; Defazio, 2025), see the Gradient Norm panel in Fig. 1. Therefore, according to (7), if the loss initially decreases and then starts to oscillate, as is often observed in training, then directional smoothness must start below $2/\eta$ and then increase (sharpen) up to $2/\eta$, and then oscillate around $2/\eta$. Indeed, see the Directional Smoothness panel in Fig. 1, where we can see that the directional smoothness progressively sharpens up to $2/\eta$. Thus, by almost definition, directional smoothness exhibits the sharpening and EoS phase.

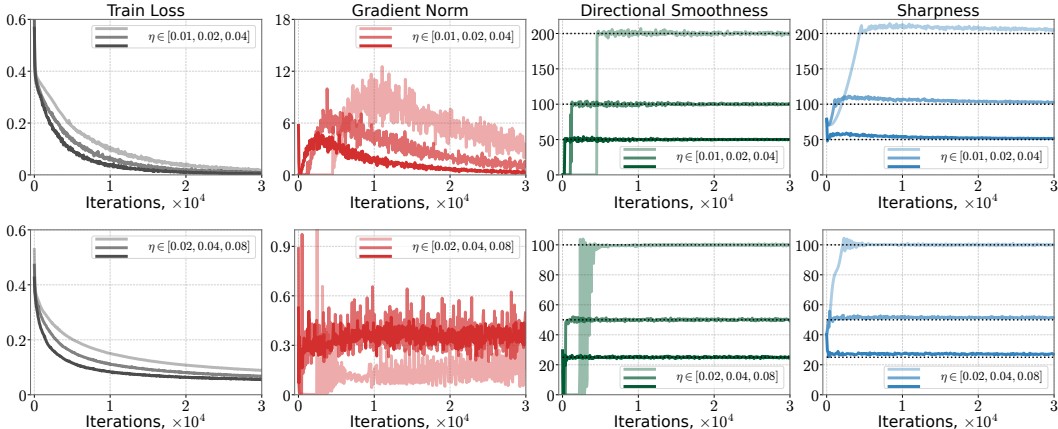

Figure 1: (Vanilla `GD`) Train loss, gradient norm, directional smoothness, and sharpness during training MLP (**top**) and CNN (**bottom**) models on CIFAR10-5k dataset with vanilla `GD`. Horizontal dashed lines correspond to the value $2/\eta$.

## 2.1 CONNECTION TO SHARPNESS

Next, we show how directional smoothness is closely related to a Hessian quantity that we will call the generalized sharpness. We can relate (5) to sharpness by using the 2nd-order Taylor expansion of our objective and one step of non-Euclidean `GD` in (1)

$$D^{\|\cdot\|}(\mathbf{w}_t, \mathbf{w}_{t+1}) := \frac{\mathcal{L}(\mathbf{w}_{t+1}) - \mathcal{L}(\mathbf{w}_t) - \langle \nabla \mathcal{L}(\mathbf{w}_t), \mathbf{w}_{t+1} - \mathbf{w}_t \rangle}{\frac{1}{2}\|\mathbf{w}_{t+1} - \mathbf{w}_t\|^2}$$

$$= \frac{\mathbf{d}_t^\top \int_{\tau=0}^1 \nabla^2 \mathcal{L}(\mathbf{w}_t - \tau\eta\mathbf{d}_t)\mathrm{d}\tau\,\mathbf{d}_t}{\|\mathbf{d}_t\|^2}. \tag{8}$$

We can further upper-bound (8) by taking the maximum over all directions

$$D^{\|\cdot\|}(\mathbf{w}_t, \mathbf{w}_{t+1}) \leq \max_{\tau \in [0,1]} \frac{\mathbf{d}_t^\top \nabla^2 \mathcal{L}(\mathbf{w}_t - \tau\eta\mathbf{d}_t)\mathbf{d}_t}{\|\mathbf{d}_t\|^2} \leq \max_{\mathbf{d}\neq 0, \tau \in [0,1]} \frac{\mathbf{d}^\top \nabla^2 \mathcal{L}(\mathbf{w}_t - \tau\eta\mathbf{d}_t)\mathbf{d}}{\|\mathbf{d}\|^2}. \tag{9}$$

If we further assume that the Hessian is almost constant over the line segment $\{\mathbf{x} : \mathbf{x} = \mathbf{w}_t - \eta\tau\mathbf{d}_t, \tau \in [0, 1]\}$, we arrive at the following definition of generalized sharpness:

**Definition 2.2.** For any norm $\|\cdot\|$, we define the *generalized sharpness* as:

$$S^{\|\cdot\|}(\mathbf{w}) := \max_{\mathbf{d}\neq 0} \frac{\mathbf{d}^\top \nabla^2 \mathcal{L}(\mathbf{w})\mathbf{d}}{\|\mathbf{d}\|^2} = \max_{\mathbf{d}} \mathbf{d}^\top \nabla^2 \mathcal{L}(\mathbf{w})\mathbf{d} \quad \text{s.t. } \|\mathbf{d}\|^2 \leq 1. \tag{10}$$

The optimization problem (10) involves *maximizing* a quadratic function over a convex constraint set, and is thus challenging to solve in general. For some choices of norm $\|\cdot\|$, the problem (10) has an analytical solution (e.g., vanilla `GD` or `Block CD`). For other norms, we will heuristically approximate the solution to (10) using the Frank-Wolfe (`FW`) algorithm (Frank et al., 1956) run from multiple random restarts (Alg. 1). On smooth, non-convex objectives, `FW` is known to converge to a first-order stationary point over convex-sets with `FW` gap as a measure (Lacoste-Julien, 2016).

Since a stationary point is not necessarily the global maximum, we repeatedly run Frank-Wolfe from multiple random restarts and then take the maximum over all trials. Empirically, we usually observe that the generalized sharpness estimated using this procedure converges to some limiting value as the number of random restarts grows. Note that in Alg. 1, we project the output of `FW` onto the unit norm sphere, as the final Frank-Wolfe iterate may lie in the interior of the norm ball while the true global maximizer

---

**Algorithm 1:** Frank-Wolfe to approximate (10)

**Input:** norm $\|\cdot\|$, $\gamma_k = \frac{2}{2+k}$, $S_0 = 0$

**for** *restart* $m = 1, \ldots, M$ **do**
    $\mathbf{u}_0 \sim \mathcal{N}(0, \boldsymbol{I})$, $\mathbf{u}_0 = \Pi_{\|\cdot\|=1}(\mathbf{u}_0)$
    **for** $k = 0, 1, \ldots, K - 1$ **do**
        $\mathbf{v}_k = \Pi_{\|\cdot\|\leq 1}(\nabla^2 \mathcal{L}(\mathbf{w}_t)\mathbf{u}_k)$
        $\mathbf{u}_{k+1} = (1 - \gamma_k)\mathbf{u}_k + \gamma_k\mathbf{v}_k$
    $\mathbf{u}_K = \Pi_{\|\cdot\|=1}(\mathbf{u}_K)$, $\hat{S}_m = \mathbf{u}_K^\top \nabla^2 \mathcal{L}(\mathbf{w}_t)\mathbf{u}_K$
    $S_m = \max\{S_{m-1}, \hat{S}_m\}$

**Output:** $S_M$

---

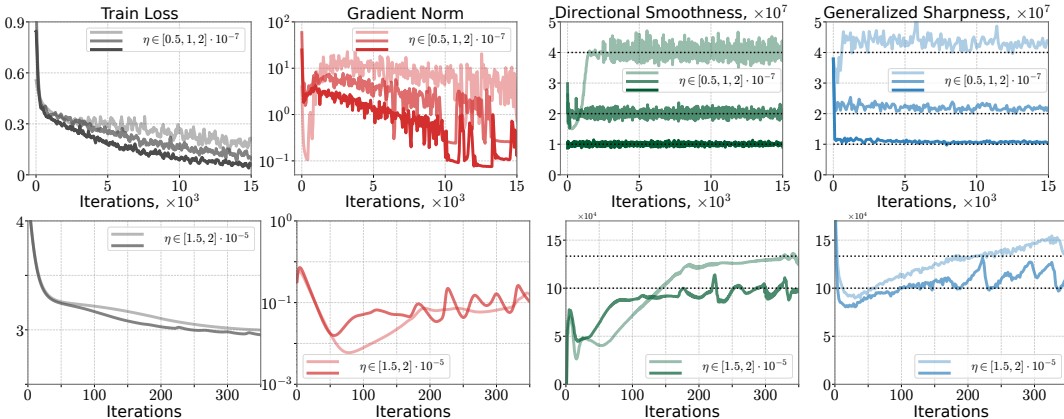

Figure 2: ($\ell_\infty$-descent) Train loss, gradient norm, directional smoothness, and generalized sharpness (14) during training MLP on CIFAR10-5k (top) and Transformer on Tiny Shakespeare (bottom) with $\ell_\infty$-descent. Horizontal dashed lines correspond to the value $2/\eta$.

must lie on the boundary. See App. A for a more detailed discussion of our procedure for approximating (10).

## 3 Examples of Non-Euclidean Gradient Descent

We begin by showing that the generalized sharpness (10) recovers previously derived notions of sharpness, establishing the tightness of our approach. We then examine generalized sharpness under several non-Euclidean norms.

**Euclidean $\ell_2$ Norm.** We consider a standard Euclidean $\ell_2$ norm. In this case, the sharpness measure (10) can be computed explicitly. Indeed, the maximum in (10) equals the largest eigenvalue of the Hessian $\lambda_{\max}(\nabla^2 \mathcal{L}(\mathbf{w}_t))$. This result coincides with the sharpness measure introduced in Cohen et al. (2021). In Fig. 1, we report the training dynamics of vanilla GD, flattening all parameters of the networks. We observe that the directional smoothness and sharpness hover at $2/\eta$ when the algorithm enters EoS stage, supporting our claims in (7).

**Preconditioned $\ell_2$ Norm.** Let $\boldsymbol{P}_t \in \mathbb{R}^{d \times d}$ be a symmetric positive definite matrix, which we will use as a preconditioner. That is, we define the preconditioned $\ell_2$ norm (also referred to as the Mahalanobis distance) by $\|\mathbf{w}\|_{\boldsymbol{P}_t}^2 := \langle \boldsymbol{P}_t \mathbf{w}, \mathbf{w} \rangle = \|\boldsymbol{P}_t^{1/2} \mathbf{w}\|_2^2$. Under this norm, preconditioned GD (1) is given by

$$\mathbf{w}_{t+1} = \mathbf{w}_t - \eta \boldsymbol{P}_t^{-1} \nabla \mathcal{L}(\mathbf{w}_t). \tag{11}$$

This case includes Adagrad (Duchi et al., 2011), RMSProp (Tieleman & Hinton, 2012) and Newton's method as special cases. According to (10), the correct notion of sharpness for this norm is given by

$$S^{\|\cdot\|_{\boldsymbol{P}_t}}(\mathbf{w}) := \max_{\mathbf{d} \neq 0} \frac{\mathbf{d}^\top \nabla^2 \mathcal{L}(\mathbf{w}) \mathbf{d}}{\|\mathbf{d}\|_{\boldsymbol{P}_t}^2} = \max_{\mathbf{v} \neq 0} \frac{\mathbf{v}^\top \boldsymbol{P}_t^{-1/2} \nabla^2 \mathcal{L}(\mathbf{w}) \boldsymbol{P}_t^{-1/2} \mathbf{v}}{\|\mathbf{v}\|_2^2}, \tag{12}$$

where we arrived at last equality by using the change of variables $\mathbf{v} = \boldsymbol{P}_t^{1/2} \mathbf{d}$. This definition matches the sharpness definition for preconditioned GD given in (Cohen et al., 2025).

**Infinity $\ell_\infty$ Norm.** In this case, we consider the infinity norm over the parameters of the neural network, that is $\|\mathbf{w}\|_\infty := \max_{j \in [d]} |\mathbf{w}_j|$. The resulting method (1) is the following variant of $\ell_\infty$-descent given by

$$\mathbf{w}_{t+1} = \mathbf{w}_t - \eta \|\nabla \mathcal{L}(\mathbf{w}_t)\|_1 \text{sign}(\nabla \mathcal{L}(\mathbf{w}_t)), \tag{13}$$

The corresponding definition of sharpness (10) under this norm is given by

$$S^{\|\cdot\|_\infty}(\mathbf{w}) = \max_{\mathbf{d} \neq 0} \frac{\mathbf{d}^\top \nabla^2 \mathcal{L}(\mathbf{w}) \mathbf{d}}{\|\mathbf{d}\|_\infty^2} = \max_{\mathbf{d}} \mathbf{d}^\top \nabla^2 \mathcal{L}(\mathbf{w}) \mathbf{d} \quad \text{s.t.} \quad \|\mathbf{d}\|_\infty \leq 1. \tag{14}$$

The optimization problem (14) has also appeared in statistical physics, where it is equivalent to finding the maximum energy—or, correspondingly, the *ground state* in a *flipped sign* formulation—of

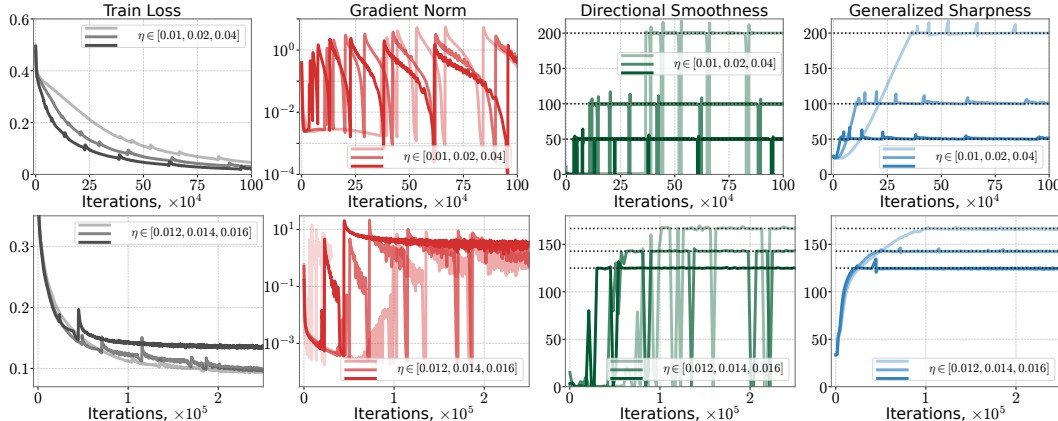

Figure 3: (`Block CD`) Train loss, gradient norm, directional smoothness, and generalized sharpness (16) during training MLP (top) and CNN (bottom) models on CIFAR10-5k dataset with `Block CD`. Horizontal dashed lines correspond to the value $2/\eta$.

an Ising spin glass on the hypercube. This corresponds to maximizing the Hamiltonian over binary spin assignments $d_i = \pm 1$. The problem is known to be NP-hard in general (Zhang & Kamenev, 2025; Kochenberger et al., 2014). Therefore, we use Alg. 1 to approximate (14), with the projection operator being $\Pi_{\|\cdot\|_\infty = 1}(\cdot) \equiv \text{sign}(\cdot)$.

Fig. 2 presents the convergence results of $\ell_\infty$-descent, applied to the flattened networks' parameters. In this case, directional smoothness plateaus at $2/\eta$. A similar behavior appears for generalized sharpness. We observe several interesting phenomena. First, in some cases, the generalized sharpness hovers *slightly above* the stability threshold $2/\eta$, . As we review in App. C, a similar effect has been observed for Euclidean GD when there are multiple Hessian eigenvalues at the edge of stability, and we hypothesize this behavior could have a similar origin. Second, FW requires a sufficient number of restarts to obtain a good approximation of the generalized sharpness in (14): see Fig. F.2.

**Block $\ell_{1,2}$ Norm.** In this case, we take into account the block-wise structure of neural networks. Let the parameters $\mathbf{w}$ be split into $L$ blocks, i.e., $\mathbf{w} = (\mathbf{w}^1, \ldots, \mathbf{w}^L) \in \mathbb{R}^{d_1} \oplus \mathbb{R}^{d_2} \ldots \oplus \mathbb{R}^{d_L}$ where $\sum_{\ell=1}^L d_\ell = d$. We consider GD in $\|\cdot\|_{1,2}$ norm[2] defined as $\|\mathbf{w}\|_{1,2} := \sum_{\ell=1}^L \|\mathbf{w}^\ell\|_2$. Let $\ell_{\max} := \text{argmax}_{\ell \in [L]} \|\nabla_{\mathbf{w}^\ell} \mathcal{L}(\mathbf{w}_t)\|$. Then GD in this norm reduces to `Block CD`

$$\mathbf{w}_{t+1}^{\ell_{\max}} = \mathbf{w}_t^{\ell_{\max}} - \eta \nabla_{\mathbf{w}^{\ell_{\max}}} \mathcal{L}(\mathbf{w}_t), \quad \mathbf{w}_{t+1}^\ell = \mathbf{w}_t^\ell \quad \text{for } \ell \neq \ell_{\max}. \tag{15}$$

The derivations of GD in this norm are given in Lemma D.5. The corresponding definition of sharpness (10) under this norm is given by

$$S^{\|\cdot\|_{1,2}}(\mathbf{w}_t) = \max_{\mathbf{d} \neq 0} \frac{\langle \mathbf{d}, \nabla^2 \mathcal{L}(\mathbf{w}_t)\mathbf{d} \rangle}{\|\mathbf{d}\|_{1,2}^2} = \max_{\mathbf{d}} \langle \mathbf{d}, \nabla^2 \mathcal{L}(\mathbf{w}_t)\mathbf{d} \rangle \quad \text{s.t.} \|\mathbf{d}\|_{1,2} \leq 1. \tag{16}$$

The solution to (16) can be given explicitly if the Hessian $\nabla^2 \mathcal{L}(\mathbf{w}_t)$ is PSD (see Lemma D.8)

$$S^{\|\cdot\|_{1,2}}(\mathbf{w}) = \max_{\ell \in [L]} \lambda_{\max}(\nabla^2_{\mathbf{w}^\ell} \mathcal{L}(\mathbf{w})). \tag{17}$$

However, for the general $\nabla^2 \mathcal{L}(\mathbf{w}_t)$, solving (16) is NP-hard (Bhattiprolu et al., 2021), but still can be approximated by the FW algorithm. The exact steps of FW in this case are derived in Lemma D.9.

Figure 3 shows the convergence of `Block CD`, where we adopt the natural block-wise structure of the network – each block corresponding to a weight matrix or bias vector of a layer. The generalized sharpness, which is approximated by the maximum eigenvalue of each block of the Hessian, approaches the threshold $2/\eta$, supporting our theoretical observations. In contrast, the directional smoothness curves display sharper dynamics: while they also reach $2/\eta$, they exhibit sudden drops whenever training shifts from a layer already at the EoS regime to one that has not yet reached it. These drops are also mirrored in the gradient norm dynamics. Similar to $\ell_\infty$, FW algorithm is sensitive to the number of restarts $M$. Fig. G.1 reports that FW with $M = 10$ provides a stable estimation of the generalized sharpness, while FW with $M = 1$ does not.

---

[2] In this case, each block $\mathbf{w}^\ell$ is treated as a vector.

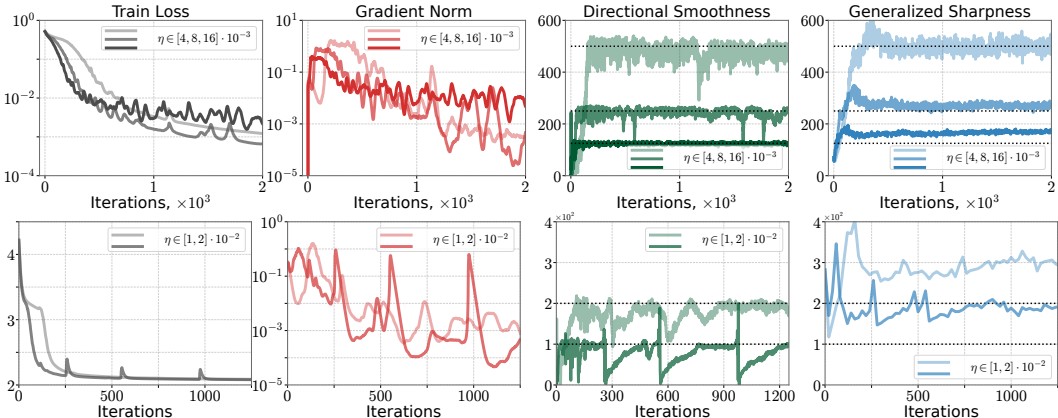

Figure 4: (`Spectral GD`) Train loss, gradient norm, directional smoothness, and generalized sharpness (19) during training MLP (top, CIFAR10) and Transformer (bottom, Tiny Shakespeare) models with the `Spectral GD`. Horizontal dashed lines correspond to the value $2/\eta$.

**Spectral $\|\cdot\|_{2\to 2}$ Norm.** To handle matrix norms, we shift perspective and treat the layers of the network as blocks of matrices[3] $\boldsymbol{W} \coloneqq (\boldsymbol{W}^1, \ldots, \boldsymbol{W}^L)$. In this setting, the natural inner product is the matrix trace $\langle \boldsymbol{W}, \boldsymbol{G} \rangle \coloneqq \mathrm{tr}\left(\boldsymbol{W}^\top \boldsymbol{G}\right)$. In this framework, one may endow each block $\boldsymbol{W}^\ell$ with a matrix norm, and then define a global norm on $\boldsymbol{W}$ by specifying an aggregation rule across layers. One particularly neat choice Bernstein & Newhouse (2024) is max over the spectral norms $\|\boldsymbol{W}\|_{\infty,2} \coloneqq \max_{\ell \in [L]} \|\boldsymbol{W}^\ell\|_2$, where $\|\boldsymbol{W}^\ell\|_2 \coloneqq \max_{\|\mathbf{d}\|_2=1} \|\boldsymbol{W}^\ell \mathbf{d}\|_2$. Under this geometry, GD aligns with the top singular directions of each layer. Concretely, the update is

$$\boldsymbol{W}_{t+1}^\ell = \boldsymbol{W}_t^\ell - \eta\gamma \boldsymbol{U}_t^\ell \boldsymbol{V}_t^\ell, \quad \gamma = \sum_{\ell=1}^L \mathrm{tr}\left(\boldsymbol{\Sigma}_t^\ell\right), \tag{18}$$

where $\boldsymbol{U}_t^\ell \boldsymbol{\Sigma}_t^\ell \boldsymbol{V}_t^\ell = \nabla_{\boldsymbol{W}^\ell} \mathcal{L}(\boldsymbol{W}_t)$ is the reduced SVD of the gradient of the $\ell$-th layer. The product $\boldsymbol{U}_t^\ell \boldsymbol{V}_t^\ell$ is also known as the polar factor of the matrix $\nabla_{\boldsymbol{W}^\ell} \mathcal{L}(\boldsymbol{W}_t)$, which can be computed efficiently on GPU using variants of the Newton-Schulz method (Jordan et al., 2024; Higham, 1986) or the `PolarExpress` (Amsel et al., 2025). The corresponding definition of sharpness (10) under this norm is given by

$$S^{\|\cdot\|_{2\to 2}}(\boldsymbol{W}) = \max_{\boldsymbol{D}\neq 0} \frac{\langle \boldsymbol{D}, \nabla^2 \mathcal{L}(\boldsymbol{W}_t)[\boldsymbol{D}]\rangle}{\|\boldsymbol{D}\|_{\infty,2}^2} = \max_{\boldsymbol{D}} \langle \boldsymbol{D}, \nabla^2 \mathcal{L}(\boldsymbol{W})[\boldsymbol{D}]\rangle \tag{19}$$

$$\text{s.t. } \|\boldsymbol{D}^\ell\|_2 \leq 1 \ \forall \ell \in [L],$$

where the operator $\nabla^2 \mathcal{L}(\boldsymbol{W})[\boldsymbol{D}]$ is the directional derivative of the gradient $\nabla^2 \mathcal{L}(\boldsymbol{W}_t)[\boldsymbol{D}] \coloneqq \frac{d}{d\epsilon} \nabla \mathcal{L}(\boldsymbol{W}_t + \epsilon\boldsymbol{D})|_{\epsilon=0}$. This is exactly the operation computed by Hessian-vector-product in PyTorch (Paszke et al., 2019). The solution to (19) cannot be computed explicitly. Therefore, we rely on the FW algorithm to approximate it. The exact steps of FW are derived in Lemma D.4.

Fig. 4 presents the convergence dynamics of `Spectral GD`. As in previous cases, both directional smoothness and generalized sharpness approach the stability threshold $2/\eta$. Notably, as with the $\ell_\infty$ norm, the generalized sharpness gradually reaches this threshold but remains slightly above it. However, in contrast to $\ell_\infty$ and $\ell_{1,2}$ norms, FW is not sensitive to the number of restarts $M$ (Fig. H.2).

## 4 NORMALIZED NON-EUCLIDEAN GRADIENT DESCENT

In this section, we demonstrate that our theoretical observations extend to normalized non-Euclidean GD. In more detail, the normalized update rule (3) with step-size $\eta$ can be rewritten as the unnormalized update rule (1) with effective step-size $\tilde{\eta} = \frac{\eta}{\|\nabla\mathcal{L}(\mathbf{w}_t)\|_*}$. Therefore, the corresponding directional smoothness $D^{\|\cdot\|}(\mathbf{w}_t, \mathbf{w}_{t+1})$ and generalized sharpness of normalized non-Euclidean GD hovers at the threshold $\frac{2}{\tilde{\eta}} = \frac{2\|\nabla\mathcal{L}(\mathbf{w}_t)\|_*}{\eta}$. This can also be derived by substituting one step of normalized non-Euclidean GD into (5), giving

$$\mathcal{L}(\mathbf{w}_{t+1}) = \mathcal{L}(\mathbf{w}_t) - \eta\left(\|\nabla\mathcal{L}(\mathbf{w}_t)\|_* - \frac{\eta}{2}D^{\|\cdot\|}(\mathbf{w}_t, \mathbf{w}_{t+1})\right). \tag{20}$$

---

[3]We use upper case notation to highlight the matrix structure.

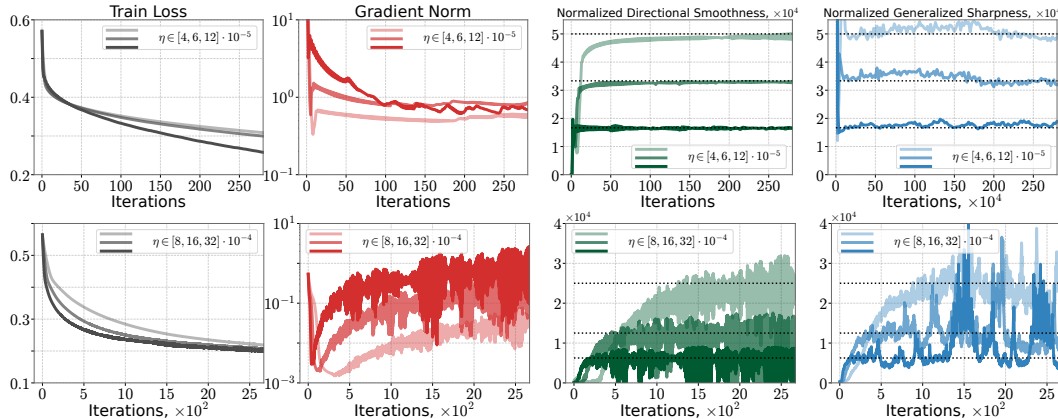

Figure 5: (Normalized non-Euclidean GD) Gradient norm, train loss, directional smoothness (normalized by the dual gradient norm), and generalized sharpness (normalized by the dual gradient norm) during training a CNN model with `SignGD` (CIFAR10-5k dataset, top line) and `Muon` without momentum (CIFAR10 dataset, bottom line). Horizontal dashed lines correspond to the value $2/\eta$.

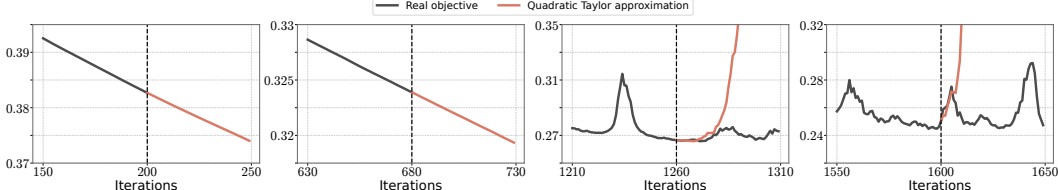

Figure 6: MSE loss ($\eta = 0.002$). At four marked iterations, we switch `Spectral GD` when training CNN on CIFAR10 from the true objective to its quadratic Taylor approximation at the current iterate (orange). (Two left, before EoS), the quadratic closely tracks the true loss; (two right, during EoS, it quickly diverges.

Therefore, the loss decreases if *and only if*

$$\mathcal{L}(\mathbf{w}_{t+1}) \leq \mathcal{L}(\mathbf{w}_t) \iff D^{\|\cdot\|}(\mathbf{w}_t, \mathbf{w}_{t+1}) \leq 2\|\nabla\mathcal{L}(\mathbf{w}_t)\|_*/\eta. \tag{21}$$

The derivations in Sec. 2.1 applies to normalized non-Euclidean GD. Fig. 5 empirically confirms the claims for `SignGD` and `Muon`, extending our EoS observations to practical algorithms. We demonstrate that the directional smoothness and generalized sharpness normalized by the dual gradient norm, i.e., $\frac{D^{\|\cdot\|}(\mathbf{w}_t, \mathbf{w}_{t+1})}{\|\nabla\mathcal{L}(\mathbf{w}_t)\|_*}$ and $\frac{S^{\|\cdot\|}(\mathbf{w}_t)}{\|\nabla\mathcal{L}(\mathbf{w}_t)\|_*}$ respectively, hover at the stability threshold $2/\eta$.

## 5 TOWARDS UNDERSTANDING THE UNDERLYING MECHANISM

For Euclidean GD, the EoS dynamics are partly understood. The significance of the sharpness $\lambda_{\max}(\nabla^2\mathcal{L}(\mathbf{w}_t))$ is that it determines whether or not GD is divergent on the local quadratic Taylor approximation. Indeed, if GD with step size $\eta$ is run on any quadratic objective function where the Hessian matrix has any eigenvalue(s) greater than $2/\eta$, then GD will oscillate with exponentially growing magnitude along the corresponding eigenvector(s). This will occur starting from almost any initialization (the one exception being if the iterate is initialized to be *exactly* orthogonal to the top eigenvector(s), an event which occurs with probability zero under any typical random initialization). Accordingly, on neural network objectives, once progressive sharpening drives the sharpness above $2/\eta$, the iterate starts to oscillate with growing magnitude along any unstable eigenvectors, just as one would expect based on the local quadratic Taylor approximation. These oscillations cause the loss to (temporarily) increase, and the directional smoothness to exceed $2/\eta$. The oscillations also crucially induce reduction of sharpness, as is revealed by considering a local cubic Taylor expansion (Damian et al., 2022), an effect which prevents the sharpness from rising further and thereby stabilizes training.

For non-Euclidean GD, since we observe that the generalized sharpness (10) (or at least, our estimate of it) hovers near $2/\eta$, it is natural to ask if an analogous explanation holds. Standard arguments from convex optimization give the following result.

**Theorem 5.1.** Let $\mathcal{L}(\mathbf{w}) \coloneqq \frac{1}{2}\mathbf{w}^\top H\mathbf{w}$ for some $H \succ 0$. For some norm $\|\cdot\|$, define the generalized sharpness $S = S^{\|\cdot\|} \coloneqq \max_{\|\mathbf{d}\| \le 1} \mathbf{d}^\top H\mathbf{d}$. If we run non-Euclidean GD (Def. 1.1) on $\mathcal{L}$ with any step-size $\eta < {}^2\!/s$, it will converge at a linear rate starting from any initial point $\mathbf{w}_0$.

See App. E for the proof. This theorem generalizes, to non-Euclidean norms, the fact that GD is convergent on quadratic functions so long as the sharpness is less than $2/\eta$. However, for the Euclidean norm, the key point is that the converse is also true: gradient descent *diverges* on quadratics if the sharpness is *greater* than $2/\eta$. We now show that this property also carries over, to an extent, to the non-Euclidean setting.

**Theorem 5.2.** Let $\mathcal{L}(\mathbf{w}) \coloneqq \frac{1}{2}\mathbf{w}^\top H\mathbf{w}$ for some $H \succ 0$. For some norm $\|\cdot\|$, define the generalized sharpness $S \coloneqq \max_{\|\mathbf{d}\| \le 1} \mathbf{d}^\top H\mathbf{d}$. If we run non-Euclidean GD (Def. 1.1) on $\mathcal{L}$, there exists an initialization $\mathbf{w}_0$ from which GD will diverge for any step-size $\eta > {}^2\!/s$.

The full proof is in App. E, and the crux is the following lemma, which implies that the direction $\hat{\mathbf{d}}$ which attains the argmax in the generalized sharpness optimization problem is an invariant direction under the non-Euclidean GD update:

**Lemma 5.3.** If $\hat{\mathbf{d}} \in \arg\max_{\|\mathbf{d}\|=1} \mathbf{d}^\top H\mathbf{d}$, then $(H\hat{\mathbf{d}})_* = \hat{\mathbf{d}}$.

As a result, if the iterate is initialized in $\mathbf{w}_0 \in \mathrm{span}(\hat{\mathbf{d}})$, then the evolution of $\mathbf{w}_t$ is given by:

$$\mathbf{w}_t = (1 - \eta S)^t \mathbf{w}_0. \tag{22}$$

When $\eta > 2/S \iff S > 2/\eta$, these dynamics oscillate with growing magnitude and diverge. However, we note that Th. 5.2 is less strong than what is true for Euclidean GD, as Euclidean GD diverges from all but a zero-measure set of initializations, whereas Th. 5.2 only establishes divergence when the initialization is on a particular line.

Empirically, we can assess whether non-Euclidean GD is indeed divergent on the quadratic Taylor approximation when operating on the edge of stability. In Fig. 6, for points during training both before and after entering EoS, we switch from running non-Euclidean GD on the real objective to running non-Euclidean GD on the quadratic Taylor approximation (similar to App. E from Cohen et al. (2021)). We observe that GD is stable before reaching EoS, but divergent afterwards. This supports the idea that the significance of the generalized sharpness hovering around $2/\eta$ is related to the dynamics becoming divergent on the local quadratic Taylor approximation.

Nevertheless, we note that our explanation of this behavior is not fully satisfying, as our theory only proves that non-Euclidean EoS is divergent under a specific initialization, whereas in practice we observe that this divergence seems to occur quite generically. Bridging this gap would be an interesting question for future work.

It is worth highlighting an additional point of difference between the Euclidean and non-Euclidean cases. For Euclidean GD, the directional smoothness only starts to grow from $\approx 0$ to $2/\eta$ *after* the sharpness crosses $2/\eta$. By contrast, for non-Euclidean GD under some norms (in particular, $\ell_\infty$ and $\|\cdot\|_{2\to 2}$), we observe that the directional smoothness starts to climb towards $2/\eta$ *before* the generalized sharpness has reached $2/\eta$ (Appendix B). During this period, we find that the iterates oscillate in weight space, but the dynamics are not yet divergent on the quadratic Taylor approximation. This suggests an intermediate regime between stability and EoS regimes, which does not occur for Euclidean GD. Understanding this behavior would be an interesting question for future work.

## 6    CONCLUSION AND FUTURE WORK

We extend EoS to previously unstudied methods such as Spectral GD, $\ell_\infty$-descent, and Muon, but several questions remain: $(i)$ the mechanism underlying stability at the $2/\eta$ threshold for general non-Euclidean GD; $(ii)$ the differing dynamics of directional smoothness in Euclidean vs. non-Euclidean GD, including a possible intermediate regime between stability and EoS; and $(iii)$ stronger convergence theory for non-Euclidean GD on quadratics, especially when $\eta > 2/S$ for arbitrary initialization.

## REPRODUCIBILITY STATEMENT

Our code base is built upon a publicly available repository (Cohen et al., 2021), incorporating necessary algorithms in the code base. All experiments utilize publicly available datasets, cited accordingly. Further details are reported in the Appendix.

## ETHICS STATEMENT

This paper presents work whose goal is to advance the field of Machine Learning. There are many potential societal consequences of our work, none of which we feel must be specifically highlighted here.

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

# Appendix

CONTENTS

## A  DISCUSSION ON FRANK-WOLFE ALGORITHM

Solving (10) reduces to the quadratic maximization problem

$$\max_{\|\mathbf{u}\|\leq 1} \mathbf{u}^\top \boldsymbol{H}\mathbf{u}, \tag{23}$$

for an arbitrary norm $\|\cdot\|$ and symmetric matrix $\boldsymbol{H}$. Even in the convex case where $\boldsymbol{H}$ is positive definite, problem (23) is NP-hard (Burer & Letchford, 2009) and is recognized as a fundamental challenge in global optimization (Horst et al., 2000). Consequently, without exploiting additional

structure, global optimality guarantees cannot be expected from generic first-order methods. Instead, one can provide stationarity-type guarantees or approximation bounds via relaxations (Burer & Letchford, 2009).

The Frank–Wolfe (FW) algorithm is a projection-free method that relies on a linear minimization oracle $\min_{\|\mathbf{w}\|=1}\langle \mathbf{w} - \mathbf{u}, \boldsymbol{H}\mathbf{u}\rangle$. For maximization problems such as (23), this oracle is applied in reverse, i.e., minimizing $-\mathbf{u}^\top \boldsymbol{H}\mathbf{u}$. For $L$-smooth functions over convex domains, which includes (23), the FW algorithm provides convergence to approximate stationary points, measured through the Frank–Wolfe gap

$$\mathcal{G}(\mathbf{u}) \coloneqq \max_{\|\mathbf{w}\|\leq 1} \langle \mathbf{w} - \mathbf{u}, -\boldsymbol{H}\mathbf{u}\rangle,$$

where the last term comes with minus since we minimize $-\mathbf{u}^\top \boldsymbol{H}\mathbf{u}$. Specifically, FW identifies an iterate $\mathbf{u}_K$ satisfying $\mathcal{G}(\mathbf{u}_K) \leq \varepsilon$ in $\mathcal{O}(1/\varepsilon^2)$ iterations, i.e., at rate $\mathcal{O}(1/\sqrt{K})$ (Lacoste-Julien, 2016). While this guarantee does not imply global optimality for (23), it provides a principled and certifiable stopping criterion. However, the solution to (23) must lie at the boundary of the unit ball in $\|\cdot\|$ norm, since the quadratic function is continuous. Therefore, in the experiments, we add a projection step. We observe that such a projection step always improved the final iterate.

As an alternative, consider the projected power iteration

$$\mathbf{u}_{k+1} = \Pi_{\|\cdot\|}(\boldsymbol{H}\mathbf{u}_k).$$

For the Euclidean norm, this reduces to the classical Power method, which converges to the normalized leading eigenvector provided the initialization has a nonzero component along it (Golub & Van Loan, 2013). For general norms, however, no global convergence guarantees are known: the projected iterates can stall or even cycle–for example, when they approach generalized eigenvectors, namely unit vectors $\mathbf{v}$ that are fixed points of the linear minimization oracle, $\mathbf{v} = \operatorname*{argmin}_{\|\mathbf{w}\|=1} \langle \mathbf{w} - \mathbf{v}, -\boldsymbol{H}\mathbf{v}\rangle$. Empirically, we found that FW provides a good estimation of (10) when a sufficient number of restarts is used.

## B  An oscillatory regime before EoS

In this appendix, we briefly elaborate on an oscillatory regime that occurs for some optimizers (including $\ell_\infty$-descent and Spectral GD) *before* the algorithm reaches EoS. This stands in contrast to Euclidean GD, which generally does not oscillate before the sharpness reaches $2/\eta$ (Cohen et al., 2024).

In Figure B.1, we train a network using $\ell_\infty$ descent. Initially, the generalized sharpness is less than $2/\eta$, the directional smoothness is $\approx 0$, and the network's predictions are not oscillating. Then, around step 300, even though the generalized sharpness is less than $2/\eta$, the directional smoothness starts to rise and the network's predictions start to oscillate, which are indications that the iterates are oscillating in weight space. Finally, around step 450, the generalized sharpness and directional smoothness reach $2/\eta$ and the algorithm reaches EoS. The network's predictions oscillate wildly.

The existence of the pre-EoS oscillatory regime is interesting, since no such regime exists for Euclidean GD.

In Figure B.2, we further explore this phenomenon. At three points during training, we switch from running $\ell_\infty$ descent on the real objective to running it on the quadratic Taylor approximation. We show the evolution of the network output under the resulting trajectory. Initially (left), the network output does not oscillate, indicating that the iterates are not oscillating in weight space. On the other hand, once the dynamics are in the pre-EoS oscillatory regime (middle), the network output oscillates but does not diverge. Finally, once the dynamics are at EoS (right), the network output diverges.

An interesting avenue for future work would be to understand why non-Euclidean GD starts to oscillate when it does.

## C  THE GAP BETWEEN THE GENERALIZED SHARPNESS AND $2/\eta$

Prior studies of Euclidean GD at EoS have observed that there is often a gap between the sharpness and $2/\eta$; for example, in Figure 1 of Cohen et al. (2021), the sharpness can be seen to sometimes exceed the critical threshold of $2/\eta$ by 150%. Similar effects can be observed in plots throughout this paper for the generalized sharpness during non-Euclidean GD. We now review the prevailing explanation for this phenomenon for Euclidean GD, and suggest that a similar mechanism is at play for non-Euclidean GD.

For Euclidean GD, Cohen et al. (2025) argue that when multiple Hessian eigenvalues are near $2/\eta$, GD should be conceived of as oscillating within the subspace spanned by the corresponding eigenvectors. The EoS phenomenon is that for every direction $\mathbf{d}$ in this subspace, the local time-average of the directional curvature $\mathbf{d}^\top \nabla^2 \mathcal{L}(\mathbf{w})\mathbf{d}$ is approximately equal to $2/\eta$. Concretely, if at some iteration $t$, one computes the top Hessian eigenvector $\mathbf{d}$, and then monitors the quantity $\mathbf{d}^\top \nabla^2 \mathcal{L}(\mathbf{w}_{t+j})\mathbf{d}$ for the next $j = 1, \dots, m$ iterations, then the local time-average of this quantity $\frac{1}{m}\sum_{j=1}^{k} \mathbf{d}^\top \nabla^2 \mathcal{L}(\mathbf{w}_{t+j})\mathbf{d}$ is predicted to be approximately $2/\eta$. By contrast, if we compute the top Hessian eigenvalue anew at every iteration $\{\lambda_{\max}(\nabla^2 \mathcal{L}(\mathbf{w}_t))\}$, then due to the chaotic oscillatory dynamics, we get back a different vector within this subspace at every step, and because the largest Hessian eigenvector is the direction with the largest curvature, there is an upward bias.

For an analogy, consider the random $d$-dimensional matrix
$$\boldsymbol{H} := \boldsymbol{U}[\tfrac{2}{\eta}\,\boldsymbol{I}_k + \varepsilon\,\mathrm{diag}(\mathbf{z})]\boldsymbol{U}^\top, \quad \mathbf{z} \sim \mathcal{N}(0, \boldsymbol{I}_k),$$
where $\boldsymbol{U} \in \mathbb{R}^{d\times k}$ has orthogonal columns and $\epsilon > 0$ is a small number. Here, $\boldsymbol{H}$ is an analogy to the Hessian, the columns of $\boldsymbol{U}$ are the $k \geq 2$ unstable Hessian eigenvectors, and the random noise $\mathbf{z}$ is an analogy to the chaotic oscillatory dynamics. The nonzero eigenvalues of $H$ are exactly $\frac{2}{\eta} + \epsilon \mathbf{z}$, and so the largest eigenvalue $\lambda_{\max}(\boldsymbol{H})$ is precisely $\frac{2}{\eta} + \epsilon \max_{1\leq i\leq k} z_i$. It can be shown that $\mathbb{E}[\max_{1\leq i\leq k} z_i] > 0$ provided that $k \geq 2$, and thus we have $\mathbb{E}[\lambda_{\max}(\boldsymbol{H})] > \frac{2}{\eta}$. On the other hand, for any fixed vector $\mathbf{v} \in \mathrm{Range}(\boldsymbol{U})$, we have that $\frac{\mathbb{E}[\mathbf{v}^\top \boldsymbol{H}\mathbf{v}]}{\|\mathbf{v}\|^2} = \frac{2}{\eta}$.

Generalizing this argument to the case of non-Euclidean GD is nontrivial, as in the non-Euclidean case we do not know if there is an analogous concept to multiple eigenvalues being at the edge of stability. Nevertheless, in Figure C.1, we empirically show that while the generalized sharpness (10) hovers strictly above $2/\eta$, if we fix a timestep $t_0$ and compute the maximizer $\mathbf{d}$ of the generalized sharpness problem (10) at this timestep, then the quadratic form $\mathbf{d}^\top \nabla^2 \mathcal{L}(\mathbf{w}_{t_0+j})\,\mathbf{d}$ computed over the next $j = 1, \dots, m$ steps is much closer to $2/\eta$.

## D  USEFUL LEMMAS

### D.1  MISSING PROOFS FOR THE SPECTRAL BLOCK NORM $\ell_{\infty,2}$

First, we derive the step of Spectral GD.

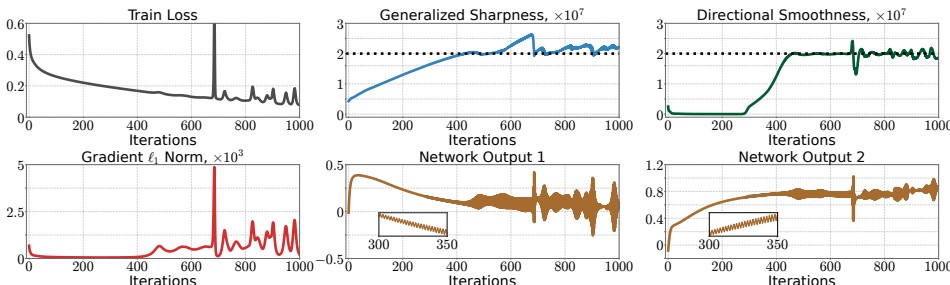

Figure B.1: **An oscillatory regime before** EoS. We train a network using $\ell_\infty$-descent. From steps $\sim 300-450$, the generalized sharpness is less than $2/\eta$ (so the algorithm is not yet at EoS), but the directional smoothness has already started to climb from $\approx 0$ towards $2/\eta$, and the network's predictions have already started to oscillate. This would not occur for Euclidean GD. This network is a fully connected network trained on a subset of CIFAR-10 using MSE loss and $\eta = 1 \times 10^{-7}$.

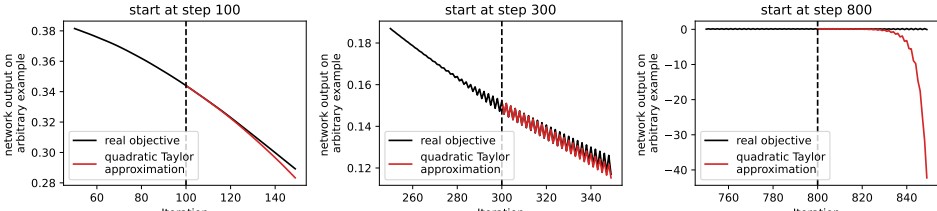

Figure B.2: **In the pre-EoS oscillatory regime, training on the quadratic Taylor approximation oscillates without diverging.** While training the network from Figure B.1, we switch from training on the real objective to to training on the quadratic Taylor approximation at three points during training: at step 100 (while the optimizer is stable and non-oscillatory), at step 300 (while the optimizer is in the pre-EoS oscillatory regime), and at step 800 (when the network is at EoS). For these trajectories, we plot the network's output on an arbitrary test example. In the first case, this output evolves smoothly; in the third case, it diverges; and, interestingly, in the second case, it oscillates with sustained magnitude and without diverging.

**Lemma D.1.** Let $\|\boldsymbol{X}^\ell\|_{\mathcal{W}_\ell}$ be the norm of the $\ell$-th layer and $\|\boldsymbol{X}\|^2 = \sum_{\ell=1}^L \|\boldsymbol{X}^\ell\|^2_{\mathcal{W}_\ell}$. The solution to

$$\boldsymbol{\Delta W}_* = \operatorname*{argmin}_{\boldsymbol{\Delta W}} \ \operatorname{tr}\left(\boldsymbol{\Delta W}^\top \boldsymbol{G}\right) + \frac{1}{2\eta}\|\boldsymbol{\Delta W}\|^2. \tag{24}$$

is given by

$$\boldsymbol{\Delta W}_*^\ell = \eta \cdot \|\boldsymbol{G}^\ell\|^*_{\mathcal{W}_\ell} \cdot \operatorname*{argmin}_{\|\boldsymbol{X}\|_{\mathcal{W}_\ell}=1} \ \operatorname{tr}\left(\boldsymbol{X}^\top \boldsymbol{G}^\ell\right) \tag{25}$$

where $\|\cdot\|^*_{\mathcal{W}_\ell}$ denotes the dual norm of $\|\cdot\|_{\mathcal{W}_\ell}$.

*Proof.* First, note that this problem is separable over each layer since

$$\operatorname{tr}\left(\boldsymbol{\Delta W}^\top \boldsymbol{G}\right) + \frac{1}{2\eta}\|\boldsymbol{W}\|^2 = \sum_{\ell=1}^L \left(\operatorname{tr}\left((\boldsymbol{\Delta W}^\ell)^\top \boldsymbol{G}^\ell\right) + \frac{1}{2\eta}\|\boldsymbol{\Delta W}^\ell\|^2_{\mathcal{W}_\ell}\right).$$

Thus, we can solve over each layer separately. Changing coordinates with $\boldsymbol{\Delta W}^\ell = c\boldsymbol{X}$ where $\|\boldsymbol{X}\|_{\mathcal{W}_\ell} = 1$ and $c \geq 0$ we have that

$$\min_{\boldsymbol{\Delta W}^\ell} \operatorname{tr}\left((\boldsymbol{\Delta W}^\ell)^\top \boldsymbol{G}^\ell\right) + \frac{1}{2\eta}\|\boldsymbol{\Delta W}^\ell\|^2_{\mathcal{W}_\ell} = \min_{c\geq 0} c \min_{\|\boldsymbol{X}\|_{\mathcal{W}_\ell}=1} \operatorname{tr}\left(\boldsymbol{X}^\top \boldsymbol{G}^\ell\right) + \frac{1}{2\eta}c^2$$

$$= \min_{c\geq 0} -c\|\boldsymbol{G}^\ell\|^*_{\mathcal{W}_\ell} + \frac{1}{2\eta}c^2.$$

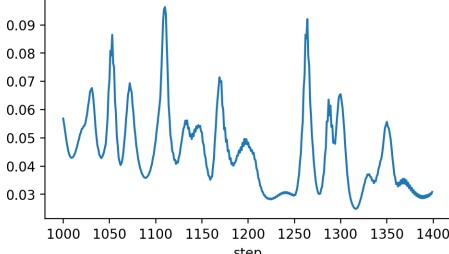
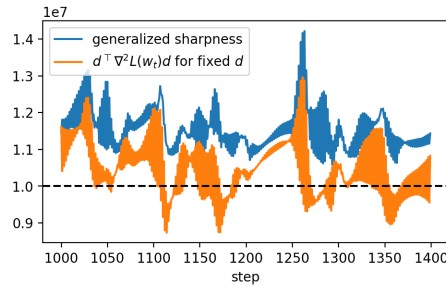

Figure C.1: For a stretch of training, we plot both the (estimated) generalized sharpness $\max_{\|\mathbf{d}\|\leq 1} \mathbf{d}^\top \nabla^2 \mathcal{L}(\mathbf{w}_t)\mathbf{d}$ (blue), as well as the quadratic form $\mathbf{d}_*^\top \nabla^2 \mathcal{L}(\mathbf{w}_t)\mathbf{d}_*$ where $\mathbf{d}_* \in \operatorname{argmax}_{\|\mathbf{d}\|\leq 1} \mathbf{d}^\top \nabla^2 \mathcal{L}(\mathbf{w}_{t_0})\mathbf{d}$ is the maximizing direction at step $t_0 = 1000$. While the first quantity is consistently larger than $2/\eta$, the second is much closer to $2/\eta$. This is a fully-connected network trained on a subset of CIFAR-10 using MSE loss and $\ell_\infty$ descent with $\eta =$2e-7.

Here, we use the fact that $\operatorname*{argmin}_{\boldsymbol{X}} \operatorname{tr}(\boldsymbol{X}^\top \boldsymbol{G}^\ell) = -(\boldsymbol{G}^\ell)^*$ is the dual matrix of $\boldsymbol{G}^\ell$. Finally solving in $c \geq 0$ gives $c = \eta \cdot \|\boldsymbol{G}_\ell\|_{\mathcal{W}_\ell}^*$.

$\square$

If we use the infinity norm over layers instead of the Euclidean one, we get the following result.

**Lemma D.2.** The solution to

$$\boldsymbol{\Delta W}_* = \operatorname*{argmin}_{\boldsymbol{\Delta W}} \operatorname{tr}\left(\boldsymbol{\Delta W}^\top \boldsymbol{G}_t\right) + \frac{1}{2\eta} \max_{\ell \in [L]} \|\boldsymbol{\Delta W}^\ell\|_{\mathcal{W}_\ell}^2. \tag{26}$$

is given by

$$\boldsymbol{\Delta W}_*^\ell = \eta\gamma \cdot \operatorname*{argmin}_{\|\boldsymbol{X}\|_{\mathcal{W}_\ell}=1} \operatorname{tr}\left(\boldsymbol{X}^\top \boldsymbol{G}_t^\ell\right) \tag{27}$$

where $\gamma := \sum_{\ell=1}^L \|\boldsymbol{G}_t^\ell\|_{\mathcal{W}_\ell}^*$ and $\|\cdot\|_{\mathcal{W}_\ell}^*$ denotes the dual norm of $\|\cdot\|_{\mathcal{W}_\ell}$.

**Remark D.3.** If $\|\cdot\|_{\mathcal{W}_\ell} = \|\cdot\|_2$ for all $\ell \in [L]$, then $\boldsymbol{\Delta W}^\ell = \eta\gamma \boldsymbol{U}_t^\ell \boldsymbol{V}_t^\ell$ where $\boldsymbol{G}_t^\ell = \boldsymbol{U}_t^\ell \boldsymbol{\Sigma}_t^\ell \boldsymbol{V}_t^\ell$ is the reduced SVD decomposition. Moreover, $\gamma = \sum_{\ell=1}^L \|\boldsymbol{G}_t^\ell\|_*$ is the sum of nuclear norms over the layers. See the proof in (Bernstein & Newhouse, 2024).

*Proof.* The problem that we want to solve is

$$\min_{\boldsymbol{\Delta W}} \sum_{\ell=1}^L \operatorname{tr}\left((\boldsymbol{\Delta W}^\ell)^\top \boldsymbol{G}_t^\ell\right) + \frac{1}{2\eta} \max_{\ell \in [L]} \|\boldsymbol{\Delta W}^\ell\|_{\mathcal{W}_\ell}^2;$$

Let $\mathcal{S} := \{\boldsymbol{\Delta W} \mid \|\boldsymbol{\Delta W}^\ell\|_{\mathcal{W}_\ell} \leq t \; \forall \ell \in [L]\}$. We can rewrite this problem as

$$\min_{t \geq 0} \min_{\boldsymbol{\Delta W} \in \mathcal{S}} \left[\sum_{\ell=1}^L \operatorname{tr}\left((\boldsymbol{\Delta W}^\ell)^\top \boldsymbol{G}_t^\ell\right) + \frac{1}{2\eta}\|\boldsymbol{\Delta W}^\ell\|_{\mathcal{W}_\ell}^2\right] = \min_{t \geq 0} \min_{\boldsymbol{\Delta W} \in \mathcal{S}} \left[\sum_{\ell=1}^L \operatorname{tr}\left((\boldsymbol{\Delta W}^\ell)^\top \boldsymbol{G}_t^\ell\right) + \frac{t^2}{2\eta}\right]$$

$$= \min_{t \geq 0} \left[\sum_{\ell=1}^L \min_{\|\boldsymbol{\Delta W}^\ell\|_{\mathcal{W}_\ell} \leq t} \operatorname{tr}\left((\boldsymbol{\Delta W}^\ell)^\top \boldsymbol{G}_t^\ell\right) + \frac{t^2}{2\eta}\right] = \min_{t \geq 0} \left[\sum_{\ell=1}^L -t \max_{\|\boldsymbol{\Delta W}^\ell\|_{\mathcal{W}_\ell} \leq 1} \operatorname{tr}\left((\boldsymbol{\Delta W}^\ell)^\top \boldsymbol{G}_t^\ell\right) + \frac{t^2}{2\eta}\right]$$

$$= \min_{t \geq 0} \left[\sum_{\ell=1}^L -t\|\boldsymbol{G}_t^\ell\|_{\mathcal{W}_\ell}^* + \frac{t^2}{2\eta}\right].$$

Now it is a quadratic problem in $t$. The minimizer $t_*$ is given by

$$t_* := \eta \sum_{\ell=1}^L \|\boldsymbol{G}_t^\ell\|_{\mathcal{W}_\ell}^*.$$

Therefore, the final solution is given by

$$\boldsymbol{\Delta W}^\ell = \eta \left(\sum_{\ell=1}^L \|\boldsymbol{G}_t^\ell\|_{\mathcal{W}_\ell}^*\right) \operatorname*{argmin}_{\|\boldsymbol{X}\|_{\mathcal{W}_\ell} \leq 1} \operatorname{tr}\left(\boldsymbol{X}^\top \boldsymbol{G}_t^\ell\right).$$

$\square$

**Lemma D.4.** Let $\|\cdot\|$ be the spectral block norm $\|\cdot\|_{2\to 2}$. Then the iterates of the FW to approximate (19) are given by

$$\boldsymbol{U}_k^\ell \boldsymbol{V}_k^\ell = \operatorname{polar}(\nabla_{\boldsymbol{W}^\ell} F(\boldsymbol{D}_t)), \quad \boldsymbol{D}_{k+1}^\ell = (1-\gamma_k)\boldsymbol{D}_k + \gamma_k \boldsymbol{U}_k^\ell \boldsymbol{V}_k^\ell,$$

where $\mathrm{polar}(\cdot)$ is the polar decomposition of a matrix, $\gamma_k = \frac{2}{2+k}$

*Proof.* We consider the Frank-Wolfe method for finding an approximate solution. For shortness, let $\boldsymbol{H} := \nabla^2 \mathcal{L}(\boldsymbol{W}_t)$, and note that the objective $F(\boldsymbol{D}) := \langle \boldsymbol{D}, \boldsymbol{H}[\boldsymbol{D}] \rangle$ is a quadratic form, whose gradient is given by

$$\nabla F(\boldsymbol{D}) = 2\boldsymbol{H}[\boldsymbol{D}].$$

To compute a step of the Frank-Wolfe method, we need to solve

$$\operatorname*{argmin}_{\boldsymbol{D}} \ \langle \nabla F(\boldsymbol{D}_k), \boldsymbol{D} \rangle \qquad \text{subject to } \|\boldsymbol{D}^\ell\|_2 \leq 1, \quad \text{for } \ell = 1, \dots, L.$$

Clearly, this problem is separable over layers and is thus equivalent to solving (Bernstein & Newhouse, 2024)

$$\boldsymbol{U}_k^\ell \boldsymbol{V}_k^\ell = \operatorname*{argmin}_{\boldsymbol{D}^\ell} \ \langle \nabla_{\boldsymbol{W}^\ell} F(\boldsymbol{D}_k), \boldsymbol{D}^\ell \rangle \qquad \text{subject to } \|\boldsymbol{D}^\ell\|_2 \leq 1,$$

where $\nabla_{\boldsymbol{W}^\ell} F(\boldsymbol{D}_k)$ is the directional derivative of the gradient of the $\ell$-th layer given by

$$\nabla_{\boldsymbol{W}^\ell} F(\boldsymbol{D}_k) = \frac{d}{d\epsilon} \nabla_{\boldsymbol{W}^\ell} \mathcal{L}(\boldsymbol{D}_k^1, \dots, \boldsymbol{D}_k^\ell + \epsilon \boldsymbol{D}^\ell, \dots, \boldsymbol{D}_k^L)\big|_{\epsilon=0}$$

and where $\boldsymbol{U}_k^\ell \boldsymbol{\Sigma}_k^\ell \boldsymbol{V}_k^\ell = \nabla_{\boldsymbol{W}^\ell} F(\boldsymbol{D}_k)$. The matrix $\boldsymbol{U}_k^\ell \boldsymbol{V}_k^\ell$ is also known as the polar factor of $\nabla_{\boldsymbol{W}^\ell} F(\boldsymbol{D}_k)$. The resulting Frank-Wolfe method is thus given by

$$\boldsymbol{U}_k^\ell \boldsymbol{V}_k^\ell = \mathrm{polar}(\nabla_{\boldsymbol{W}^\ell} F(\boldsymbol{D}_k)), \quad \boldsymbol{D}_{k+1}^\ell = (1 - \gamma_k)\boldsymbol{D}_k^\ell + \gamma_k \boldsymbol{U}_k^\ell \boldsymbol{V}_k^\ell,$$

where $\gamma_k = \frac{2}{k+2}$. $\qquad\square$

### D.2 Missing Proofs for the Block $\ell_{1,2}$ Norm

**Lemma D.5.** The solution to the problem

$$\Delta \mathbf{w}_* = \operatorname*{argmin}_{\mathbf{w}} \ \langle \Delta \mathbf{w}, \mathbf{g}_t \rangle + \frac{1}{2\eta} \|\Delta \mathbf{w}\|_{1,2}^2$$

can be written as

$$\Delta \mathbf{w}_*^\ell = \begin{cases} 0 & \text{if } \mathbf{g}_t = 0, \\ 0 & \text{if } \mathbf{g}_t \neq 0 \text{ and } \ell \notin J, \\ -\frac{\eta}{|J|} \mathbf{g}_t^\ell & \ell \in J, \end{cases}$$

where $J := \{\ell \in [L] \mid \|\mathbf{g}_t^\ell\|_2 = \max_{j \in [L]} \|\mathbf{g}_t^j\|_2\}$.

**Remark D.6.** In the case when $J$ is a singleton, we obtain `Block CD`

$$\mathbf{w}_{t+1}^\ell = \begin{cases} \mathbf{w}_t^\ell - \eta \mathbf{g}_t^\ell & \text{if } \ell = \ell_{\max}, \\ \mathbf{w}_t^\ell & \text{otherwise}, \end{cases}$$

where $\ell_{\max} = \operatorname*{argmax}_{\ell \in [L]} \|\mathbf{g}_t^\ell\|_2$.

**Remark D.7.** In the case when $L = d$, we obtain vanilla coordinate descent (`CD`)

$$\mathbf{w}_{t+1}^j = \begin{cases} \mathbf{w}_t^{j_{\max}} - \eta \mathbf{g}_t^{j_{\max}} & \text{if } j = j_{\max} \\ \mathbf{w}_t^j & \text{otherwise}, \end{cases}$$

where $j_{\max} = \operatorname*{argmax}_{j \in [d]} |\mathbf{g}_t^j|$.

*Proof.* We need to find a solution to the problem

$$\min_{\Delta\mathbf{w}}\langle\Delta\mathbf{w},\mathbf{g}_t\rangle+\frac{1}{2\eta}\left(\sum_{\ell=1}^{L}\|\Delta\mathbf{w}^\ell\|_2\right)^2=\min_{\Delta\mathbf{w}}\sum_{\ell=1}^{L}\langle\Delta\mathbf{w}^\ell,\mathbf{g}_t^\ell\rangle+\frac{1}{2\eta}\left(\sum_{\ell=1}^{L}\|\Delta\mathbf{w}^\ell\|_2\right)^2$$

Let $\Delta\mathbf{w}_*$ be the solution to the problem. Therefore,

$$0\in\mathbf{g}_t+\frac{1}{\eta}\left(\sum_{\ell=1}^{L}\|\Delta\mathbf{w}_*^\ell\|_2\right)\partial\left(\sum_{\ell=1}^{L}\|\Delta\mathbf{w}_*^\ell\|_2\right)$$

$$=\mathbf{g}_t+\frac{1}{\eta}\left(\sum_{\ell=1}^{L}\|\Delta\mathbf{w}_*^\ell\|_2\right)(\partial\|\Delta\mathbf{w}_*^1\|_2^\top,\ldots,\partial\|\Delta\mathbf{w}_*^L\|_2^\top)^\top. \tag{28}$$

Let $\chi=\sum_{\ell=1}^{L}\|\Delta\mathbf{w}_*^\ell\|_2$. Note that

$$\partial\|\mathbf{x}\|=\begin{cases}\frac{\mathbf{x}}{\|\mathbf{x}\|_2}&\text{if }\mathbf{x}\neq0,\\\{\mathbf{y}\mid\|\mathbf{y}\|_2\leq1\}&\text{otherwise}\end{cases}.$$

Therefore, we should satisfy the following $L$ equalities

$$-\mathbf{g}_t^\ell=\frac{\chi}{\eta}\partial\|\Delta\mathbf{w}_*^\ell\|_2,\quad\text{and}\quad\|\mathbf{g}_t^\ell\|_2=\frac{\chi}{\eta}\left\|\partial\|\Delta\mathbf{w}_*^\ell\|_2\right\|\leq\frac{\chi}{\eta}. \tag{29}$$

This implies that each block of $\mathbf{g}_t$ has a norm at most $\chi/\eta$, and whenever some block $\ell$ satisfies $\partial\|\Delta\mathbf{w}_*^\ell\|_2=\frac{\Delta\mathbf{w}_*^\ell}{\|\Delta\mathbf{w}_*^\ell\|_2}$, then the corresponding block $\|\mathbf{g}_t^\ell\|_2=\frac{\chi}{\eta}$.

If $\|\mathbf{g}_t^\ell\|_2=0$ for all $\ell\in[L]$, i.e., $\mathbf{g}_t=0$, then for all $\Delta\mathbf{w}_*^\ell=0$.

Now let us assume that there is at least one block $\ell\in[L]$ such that $\|\mathbf{g}_t^\ell\|_2\neq0$. Let $J:=\{\ell\in[L]\mid\|\mathbf{g}_t^\ell\|_2=\max_{j\in[L]}\|\mathbf{g}_t^j\|_2\}\neq\emptyset$. Then, for all blocks $\ell\in J$ we have $\|\mathbf{g}_t^\ell\|_2=\frac{\chi}{\eta}$. Indeed, if it is not the case, i.e., if for all $\ell\in[L]$ we have $\|\mathbf{g}_t^\ell\|_2<\frac{\chi}{\eta}$, then $\Delta\mathbf{w}^*=0$ and we obtain a contradiction to (28) since $\mathbf{g}_t\neq0$.

We summarize that for any block $\ell\notin J$ such that $\|\mathbf{g}^\ell\|_2<\frac{\chi}{\eta}$ we obtain $\Delta\mathbf{w}_*^\ell=0$. In the opposite case for $\ell\in J$, we have that

$$\|\mathbf{g}_t^\ell\|_2=\max_{j\in[L]}\|\mathbf{g}_t^j\|_2=\frac{\chi}{\eta}\Rightarrow\chi=\sum_{\ell\in J}\|\Delta\mathbf{w}_*^\ell\|_2=|J|\max_{\ell\in J}\|\Delta\mathbf{w}_*^\ell\|=\eta\max_{\ell\in[L]}\|\mathbf{g}_t^\ell\|,$$

and from (29) we obtain $\Delta\mathbf{w}_*^\ell=-\frac{\eta\max_{j\in[L]}\|\mathbf{g}_t^j\|_2}{|J|}\frac{\mathbf{g}_t^\ell}{\|\mathbf{g}_t^\ell\|_2}=-\frac{\eta}{|J|}\mathbf{g}_t^\ell$ for $\ell\in J$. This concludes the proof. $\square$

---

**Lemma D.8.** Let $\|\cdot\|$ be the block $\ell_{1,2}$ norm. Assume that the Hessian $\nabla^2\mathcal{L}(\mathbf{w}_t)$ is positive semidefinite. Then the generalized sharpness (16) is given by

$$S^{\|\cdot\|_{1,2}}(\mathbf{w}_t)=\max_{\ell\in[L]}\lambda_{\max}(\nabla_{\mathbf{w}^\ell}^2(\mathbf{w}_t)).$$

---

*Proof.* If $\boldsymbol{H}=\nabla^2\mathcal{L}(\mathbf{w}_t)$ is positive semidefinite, then the function $f(\mathbf{d})=\langle\mathbf{d},\boldsymbol{H}\mathbf{d}\rangle$ is convex. Our goal is to find the maximum of this quadratic convex function over a $\ell_{1,2}$-norm unit ball. It attains the maximum at the border, i.e., $\|\mathbf{d}\|_{1,2}=1$. Any point $\mathbf{y}$ at the border of the $\ell_{1,2}$ unit norm can be expressed as

$$\mathbf{y}=(\alpha_1\mathbf{d}^1,\ldots,\alpha_L\mathbf{d}^L)\quad\text{where}\quad\|\mathbf{d}^\ell\|_2=1\;\forall\ell\in[L]\quad\text{and}\quad\sum_{\ell=1}^{L}\alpha_\ell=1.$$

Let $\mathbf{y}_1=(\mathbf{d}^1,0,\ldots,0),\mathbf{y}_2=(0,\mathbf{d}^2,\ldots,0),\ldots,\mathbf{y}_L=(0,0,\ldots,\mathbf{d}^L)$, $\|\mathbf{d}^\ell\|_2=1$ for all $\ell\in[L]$. Then $\mathbf{y}=\sum_{\ell=1}^{L}\alpha_\ell\mathbf{y}_\ell$. Since $f$ is convex, then $f(\mathbf{y})\leq\sum_{\ell=1}^{L}\alpha_\ell f(\mathbf{y}_\ell)\leq\max_{\ell\in[L]}f(\mathbf{y}_\ell)$. Therefore, our problem reduces to

$$\max_{\ell\in[L]}\max_{\|\mathbf{d}^\ell\|_2=1}\langle\mathbf{d}^\ell,\nabla_{\mathbf{w}^\ell}^2\mathcal{L}(\mathbf{w}_t)\mathbf{d}^\ell\rangle=\max_{\ell\in[L]}\lambda_{\max}(\nabla_{\mathbf{w}^\ell}^2\mathcal{L}(\mathbf{w}_t)), \tag{30}$$

where $\nabla^2_{\mathbf{w}^\ell}\mathcal{L}(\mathbf{w}_t)$ is the $\ell$-th diagonal block of the Hessian. In the special case of $L = d$, we have the sharpness measure

$$\max_{\mathbf{d}} \frac{\mathbf{d}^\top \nabla^2 \mathcal{L}(\mathbf{w}_t) \mathbf{d}}{\|\mathbf{d}\|_1^2} = \max_j |\nabla^2 \mathcal{L}(\mathbf{w}_t)_{jj}|.$$

$\square$

**Lemma D.9.** Let $\|\cdot\|$ be the block $\ell_{1,2}$ norm. Then the iterates of the FW to approximate (16) are given by

$$\mathbf{v}_k = \frac{(\nabla^2 \mathcal{L}(\mathbf{w}_t)\mathbf{d}_k)_\ell}{\|(\nabla^2 \mathcal{L}(\mathbf{w}_t)\mathbf{d}_k)_\ell\|_2}, \quad \mathbf{d}_{k+1} = (1 - \gamma_k)\mathbf{d}_k + \gamma_k \mathbf{v}_k,$$

where $(\nabla^2 \mathcal{L}(\mathbf{w}_t)\mathbf{d}_k)_\ell$ is the $\ell$-th block of the vector $\nabla^2 \mathcal{L}(\mathbf{w}_t)\mathbf{d}_k$, and $\gamma_k = \frac{2}{2+k}$.

*Proof.* We consider the Frank-Wolfe method for finding an approximate solution. For shortness, let $\mathbf{H} := \nabla^2 \mathcal{L}(\mathbf{w}_t)$, and not that the objective $F(\mathbf{d}) := \mathbf{d}^\top \mathbf{H}\mathbf{d}$ is a quadratic form, whose gradient is given by $\nabla F(\mathbf{d}) = 2\mathbf{H}\mathbf{d}$. To compute a step of the Frank-Wolfe method, we need to solve

$$\operatorname*{argmin}_{\mathbf{d}} \langle \nabla F(\mathbf{d}_k), \mathbf{d} \rangle \quad \text{subject to } \|\mathbf{d}\|_{1,2} \leq 1.$$

The solution to this is given by the dual norm and the dual gradient

$$\min_{\|\mathbf{d}\|_{1,2} \leq 1} \langle \nabla F(\mathbf{d}_k), \mathbf{d} \rangle = \|\nabla F(\mathbf{d}_k)\|_{\infty,2} = \max_{\ell \in [L]} \|\nabla_{\mathbf{d}^\ell} F(\mathbf{d}_k)\|_2.$$

This is true, since

$$\langle \nabla F(\mathbf{d}_k), \mathbf{d} \rangle = \sum_{\ell=1}^L \langle \nabla_{\mathbf{d}^\ell} F(\mathbf{d}_k), \mathbf{d}^\ell \rangle \leq \sum_{\ell=1}^L \|\nabla_{\mathbf{d}^\ell} F(\mathbf{d}_k)\|_2 \cdot \|\mathbf{d}^\ell\|_2$$

$$\leq \max_{\ell \in [L]} \|\nabla_{\mathbf{d}^\ell} F(\mathbf{d}_k)\|_2 \cdot \sum_{\ell=1}^L \|\mathbf{d}^\ell\|_2 = \max_{\ell \in [L]} \|\nabla_{\mathbf{d}^\ell} F(\mathbf{d}_k)\|_2. \tag{31}$$

The maximizer is obtained by concentrating all mass on any group $\ell \in \{\ell : \|\nabla_{\mathbf{d}^\ell} F(\mathbf{d}_k)\|_2 = \max_{i \in [L]} \|\nabla_{\mathbf{d}^i} F(\mathbf{d}_k)\|_2\}$, namely,

$$\mathbf{d}_*^\ell = \begin{cases} \frac{\nabla_{\mathbf{d}^\ell} F(\mathbf{d}_k)}{\|\nabla_{\mathbf{d}^\ell} F(\mathbf{d}_k)\|_2}, & \ell \in \{j : \|\nabla_{\mathbf{d}^j} F(\mathbf{d}_k)\|_2 = \max_{i \in [L]} \|\nabla_{\mathbf{d}^i} F(\mathbf{d}_k)\|_2\} \\ 0, & \text{otherwise.} \end{cases}$$

$\square$

# E    NON-EUCLIDEAN GRADIENT DESCENT ON QUADRATICS

To prove convergence of Non-Euclidean GD for the case of a sufficiently small step size, (Theorem 5.1) we follow standard arguments of smoothness and strong convexity. The following definitions of smoothness and strong convexity are standard generalizations from the Euclidean norm to an arbitrary norm.

**Definition E.1.** We say that $\mathcal{L} : \mathbb{R}^d \to \mathbb{R}$ is $(L, \|\cdot\|)$-smooth if

$$\|\nabla\mathcal{L}(\mathbf{w}) - \nabla\mathcal{L}(\mathbf{v})\|_* \leq L\|\mathbf{w} - \mathbf{v}\| \tag{32}$$

for all $\mathbf{w}, \mathbf{v} \in \mathbb{R}^d$.

**Definition E.2.** We say that $\mathcal{L} : \mathbb{R}^d \to \mathbb{R}$ is $(\mu, \|\cdot\|)$-strongly convex if

$$\mathcal{L}(\mathbf{v}) \geq \mathcal{L}(\mathbf{w}) + \langle \nabla\mathcal{L}(\mathbf{w}), \mathbf{v} - \mathbf{w} \rangle + \frac{\mu}{2}\|\mathbf{v} - \mathbf{w}\|^2 \tag{33}$$

for all $\mathbf{w}, \mathbf{v} \in \mathbb{R}^d$.

The following lemmas show that our quadratic $\mathcal{L}(\mathbf{w}) = \frac{1}{2}\mathbf{w}^\top \boldsymbol{H}\mathbf{w}$ is smooth and strongly convex.

**Lemma E.3.** The objective $\mathcal{L}(\mathbf{w}) = \frac{1}{2}\mathbf{w}^T \boldsymbol{H}\mathbf{w}$ is $(L, \|\cdot\|)$-smooth with $L = \sup_{\|\mathbf{z}\|=1} \mathbf{z}^T \boldsymbol{H}\mathbf{z}$.

*Proof.* For any $\mathbf{w}, \mathbf{v} \in \mathbb{R}^d$, denote $\mathbf{d} = (\mathbf{w} - \mathbf{v})/\|\mathbf{w} - \mathbf{v}\|$. Then

$$\frac{\|\nabla\mathcal{L}(\mathbf{w}) - \nabla\mathcal{L}(\mathbf{v})\|_*}{\|\mathbf{w} - \mathbf{v}\|} = \frac{\|\boldsymbol{H}\mathbf{w} - \boldsymbol{H}\mathbf{v}\|_*}{\|\mathbf{w} - \mathbf{v}\|} = \|\boldsymbol{H}\mathbf{d}\|_* = \sup_{\|\mathbf{u}_1\|=1} \mathbf{u}_1^\top \boldsymbol{H}\mathbf{d} \leq \sup_{\|\mathbf{u}_1\|=\|\mathbf{u}_2\|=1} \mathbf{u}_1^\top \boldsymbol{H}\mathbf{u}_2, \tag{34}$$

where in the third equality we used the definition of dual norm. Next we will prove that

$$\sup_{\|\mathbf{u}_1\|=\|\mathbf{u}_2\|=1} \mathbf{u}_1^\top \boldsymbol{H}\mathbf{u}_2 = \sup_{\|\mathbf{z}\|=1} \mathbf{z}^\top \boldsymbol{H}\mathbf{z}.$$

The $(\geq)$ direction is immediate since

$$\sup_{\|\mathbf{u}_1\|=\|\mathbf{u}_2\|=1} \mathbf{u}_1^\top \boldsymbol{H}\mathbf{u}_2 \geq \sup_{\|\mathbf{z}\|=1} \mathbf{z}^\top \boldsymbol{H}\mathbf{z}. \tag{35}$$

To show the other direction, let

$$(\mathbf{u}_1^*, \mathbf{u}_2^*) \in \operatorname*{argmax}_{\|\mathbf{u}_1\|=\|\mathbf{u}_2\|=1} \mathbf{u}_1^\top \boldsymbol{H}\mathbf{u}_2, \tag{36}$$

and

$$\mathbf{z}^* \in \operatorname*{argmax}_{\|\mathbf{z}\|=1} \mathbf{z}^\top \boldsymbol{H}\mathbf{z}. \tag{37}$$

Note that these $\operatorname{argmax}$ operations make sense, since we are considering the maximum of continuous functions on compact domains, which always achieve their supremum. Then

$$(\mathbf{u}_1^* - \mathbf{u}_2^*)^\top \boldsymbol{H}(\mathbf{u}_1^* - \mathbf{u}_2^*) \geq 0$$
$$(\mathbf{u}_1^*)^\top \boldsymbol{H}\mathbf{u}_1^* - 2(\mathbf{u}_1^*)^\top \boldsymbol{H}\mathbf{u}_2^* + (\mathbf{u}_2^*)^\top \boldsymbol{H}\mathbf{u}_2^* \geq 0$$
$$(\mathbf{u}_1^*)^\top \boldsymbol{H}\mathbf{u}_1^* + (\mathbf{u}_2^*)^\top \boldsymbol{H}\mathbf{u}_2^* \geq 2(\mathbf{u}_1^*)^\top \boldsymbol{H}\mathbf{u}_2^*$$
$$2(\mathbf{z}^*)^\top \boldsymbol{H}\mathbf{z}^* \geq 2(\mathbf{u}_1^*)^\top \boldsymbol{H}\mathbf{u}_2^*$$
$$(\mathbf{z}^*)^\top \boldsymbol{H}\mathbf{z}^* \geq (\mathbf{u}_1^*)^\top \boldsymbol{H}\mathbf{u}_2^*,$$

where the first inequality uses that $\boldsymbol{H}$ is PSD, the second inequality uses that $\boldsymbol{H}$ is symmetric, and the fourth inequality uses $(\mathbf{u}_1^*)^\top \boldsymbol{H}\mathbf{u}_1^* \leq (\mathbf{z}^*)^\top \boldsymbol{H}\mathbf{z}^*$ and $(\mathbf{u}_2^*)^\top \boldsymbol{H}\mathbf{u}_2^* \leq (\mathbf{z}^*)^\top \boldsymbol{H}\mathbf{z}^*$. This proves the $(\leq)$ direction, and proves the claim. Then Equation (34) becomes

$$\frac{\|\nabla\mathcal{L}(\mathbf{w}) - \nabla\mathcal{L}(\mathbf{v})\|_*}{\|\mathbf{w} - \mathbf{v}\|} \leq \sup_{\|\mathbf{z}\|=1} \mathbf{z}^\top \boldsymbol{H}\mathbf{z}, \tag{38}$$

or

$$\|\nabla\mathcal{L}(\mathbf{w}) - \nabla\mathcal{L}(\mathbf{v})\|_* \leq \left(\sup_{\|\mathbf{z}\|=1} \mathbf{z}^\top \boldsymbol{H}\mathbf{z}\right)\|\mathbf{w} - \mathbf{v}\|. \tag{39}$$

$\square$

**Lemma E.4.** The objective $\mathcal{L}(\mathbf{w}) = \frac{1}{2}\mathbf{w}^T \boldsymbol{H}\mathbf{w}$ is $(\mu, \|\cdot\|)$-strongly convex with $\mu = \inf_{\|\mathbf{v}\|=1} \mathbf{v}^T \boldsymbol{H}\mathbf{v}$.

*Proof.* The strong convexity property

$$\mathcal{L}(\mathbf{v}) \geq \mathcal{L}(\mathbf{w}) + \langle \nabla\mathcal{L}(\mathbf{w}), \mathbf{v} - \mathbf{w}\rangle + \frac{\mu}{2}\|\mathbf{v} - \mathbf{w}\|^2 \tag{40}$$

for our particular $\mathcal{L}$ is equivalent to each of the following statements:

$$\frac{1}{2}\mathbf{v}^\top \boldsymbol{H}\mathbf{v} \geq \frac{1}{2}\mathbf{w}^\top \boldsymbol{H}\mathbf{w} + (\mathbf{v} - \mathbf{w})^\top \boldsymbol{H}\mathbf{w} + \frac{\mu}{2}\|\mathbf{v} - \mathbf{w}\|^2 \tag{41}$$

$$\frac{1}{2}\mathbf{v}^\top \boldsymbol{H}\mathbf{v} - \mathbf{v}^\top \boldsymbol{H}\mathbf{w} + \frac{1}{2}\mathbf{w}^\top \boldsymbol{H}\mathbf{w} \geq \frac{\mu}{2}\|\mathbf{v} - \mathbf{w}\|^2 \tag{42}$$

$$(\mathbf{v} - \mathbf{w})^\top \boldsymbol{H}(\mathbf{v} - \mathbf{w}) \geq \mu\|\mathbf{v} - \mathbf{w}\|^2 \tag{43}$$

$$\left(\frac{\mathbf{v} - \mathbf{w}}{\|\mathbf{v} - \mathbf{w}\|}\right)^\top \boldsymbol{H}\frac{\mathbf{v} - \mathbf{w}}{\|\mathbf{v} - \mathbf{w}\|} \geq \mu, \tag{44}$$

which is satisfied by $\mu = \inf_{\|\mathbf{v}\|=1} \mathbf{v}^T \boldsymbol{H}\mathbf{v}$. $\qquad\square$

**Theorem 5.1.** Let $\mathcal{L}(\mathbf{w}) := \frac{1}{2}\mathbf{w}^\top \boldsymbol{H}\mathbf{w}$ for some $\boldsymbol{H} \succ 0$. For some norm $\|\cdot\|$, define the generalized sharpness $S = S^{\|\cdot\|} := \max_{\|\mathbf{d}\|\leq 1} \mathbf{d}^\top \boldsymbol{H}\mathbf{d}$. If we run non-Euclidean GD (Def. 1.1) on $\mathcal{L}$ with any step-size $\eta < 2/S$, it will converge at a linear rate starting from any initial point $\mathbf{w}_0$.

*Proof.* To show convergence, we prove a generalization of the Polyak-Łojasiewicz (PL) property, then follow the standard analysis of gradient descent for smooth and PL functions.

Lemma E.4 implies that $\mathcal{L}$ is $\mu$-strongly convex with $\mu = \inf_{\|\mathbf{v}\|=1} \mathbf{v}^\top \boldsymbol{H}\mathbf{v}$. We also know that $\mathcal{L}(\mathbf{w}) \geq \mathcal{L}_* := 0$, and that this minimum is achieved at $\mathbf{w}_* = \mathbf{0}$. So we apply (33) with $\mathbf{v} = \mathbf{w}_*$ and any $\mathbf{w}$:

$$\mathcal{L}_* \geq \mathcal{L}(\mathbf{w}) + \langle \nabla \mathcal{L}(\mathbf{w}), \mathbf{w}_* - \mathbf{w}\rangle + \frac{\mu}{2}\|\mathbf{w}_* - \mathbf{w}\|^2 \tag{45}$$

$$\geq \inf_{\mathbf{v}}\left\{\mathcal{L}(\mathbf{w}) + \langle \nabla\mathcal{L}(\mathbf{w}), \mathbf{v} - \mathbf{w}\rangle + \frac{\mu}{2}\|\mathbf{v} - \mathbf{w}\|^2\right\}. \tag{46}$$

From (1), we know the inf above is minimized when $\mathbf{v} = \mathbf{w} - 1/\mu\|\nabla\mathcal{L}(\mathbf{w})\|_*(\nabla\mathcal{L}(\mathbf{w}))_*$. We also know that $\mathcal{L}(\mathbf{w}) \geq \mathcal{L}_* := 0$ for all $\mathbf{w}$. So

$$\mathcal{L}_* \geq \mathcal{L}(\mathbf{w}) - \frac{1}{\mu}\|\nabla\mathcal{L}(\mathbf{w})\|_*\langle\nabla\mathcal{L}(\mathbf{w}), (\nabla\mathcal{L}(\mathbf{w}))_*\rangle + \frac{1}{2\mu}\|\nabla\mathcal{L}(\mathbf{w})\|_*^2\|(\nabla\mathcal{L}(\mathbf{w}))_*\|^2 \tag{47}$$

$$= \mathcal{L}(\mathbf{w}) - \frac{1}{\mu}\|\nabla\mathcal{L}(\mathbf{w})\|_*^2 + \frac{1}{2\mu}\|\nabla\mathcal{L}(\mathbf{w})\|_*^2 \tag{48}$$

$$= \mathcal{L}(\mathbf{w}) - \frac{1}{2\mu}\|\nabla\mathcal{L}(\mathbf{w})\|_*^2, \tag{49}$$

so

$$\|\nabla\mathcal{L}(\mathbf{w})\|_*^2 \geq 2\mu(\mathcal{L}(\mathbf{w}) - \mathcal{L}_*), \tag{50}$$

which is the PL property we need.

Lemma E.3 implies that $\mathcal{L}$ is $L$-smooth with $L = S$, so

$$\mathcal{L}(\mathbf{w}_{t+1}) \leq \mathcal{L}(\mathbf{w}_t) + \langle\nabla\mathcal{L}(\mathbf{w}_t), \mathbf{w}_{t+1} - \mathbf{w}_t\rangle + \frac{S}{2}\|\mathbf{w}_{t+1} - \mathbf{w}_t\|^2 \tag{51}$$

$$\leq \mathcal{L}(\mathbf{w}_t) + \eta\|\nabla\mathcal{L}(\mathbf{w}_t)\|_*\langle\nabla\mathcal{L}(\mathbf{w}_t), (\nabla\mathcal{L}(\mathbf{w}_t))_*\rangle + \frac{S\eta^2\|\nabla\mathcal{L}(\mathbf{w}_t)\|_*^2}{2}\|(\nabla\mathcal{L}(\mathbf{w}_t))_*\|^2 \tag{52}$$

$$\leq \mathcal{L}(\mathbf{w}_t) - \eta\|\nabla\mathcal{L}(\mathbf{w}_t)\|_*^2 + \frac{S\eta^2\|\nabla\mathcal{L}(\mathbf{w}_t)\|_*^2}{2} \tag{53}$$

$$\leq \mathcal{L}(\mathbf{w}_t) - \eta\left(1 - \frac{\eta S}{2}\right)\|\nabla\mathcal{L}(\mathbf{w}_t)\|_*^2 \tag{54}$$

$$\leq \mathcal{L}(\mathbf{w}_t) - 2\mu\eta\left(1 - \frac{\eta S}{2}\right)(\mathcal{L}(\mathbf{w}_t) - \mathcal{L}_*), \tag{55}$$

where the last line uses the PL property from (50) and that $\eta < 2/S$. Subtracting $\mathcal{L}_*$ from both sides:

$$\mathcal{L}(\mathbf{w}_{t+1}) - \mathcal{L}_* \leq \left(1 - 2\mu\eta\left(1 - \frac{\eta S}{2}\right)\right)(\mathcal{L}(\mathbf{w}_t) - \mathcal{L}_*), \tag{56}$$

so that for all $t$,

$$\mathcal{L}(\mathbf{w}_t) - \mathcal{L}_* \leq \left(1 - 2\mu\eta\left(1 - \frac{\eta S}{2}\right)\right)^t (\mathcal{L}(\mathbf{w}_0) - \mathcal{L}_*). \tag{57}$$

$\square$

The key to showing divergence when $\eta > 2/S$ (Theorem 5.2) is the following lemma.

**Lemma 5.3.** If $\hat{\mathbf{d}} \in \arg\max_{\|\mathbf{d}\|=1} \mathbf{d}^\top \mathbf{H} \mathbf{d}$, then $(\mathbf{H}\hat{\mathbf{d}})_* = \hat{\mathbf{d}}$.

*Proof.* Since $\mathbf{H}$ is symmetric and PSD, we have for any such $\mathbf{v}$

$$(\mathbf{v} - \hat{\mathbf{w}})^\intercal \mathbf{H}(\mathbf{v} - \hat{\mathbf{w}}) \geq 0 \tag{58}$$

$$\mathbf{v}^\intercal \mathbf{H}\mathbf{v} - 2\mathbf{v}^\intercal \mathbf{H}\hat{\mathbf{w}} + \hat{\mathbf{w}}^\intercal \mathbf{H}\hat{\mathbf{w}} \geq 0 \tag{59}$$

$$\mathbf{v}^\intercal \mathbf{H}\mathbf{v} + \hat{\mathbf{w}}^\intercal \mathbf{H}\hat{\mathbf{w}} \geq 2\mathbf{v}^\intercal \mathbf{H}\hat{\mathbf{w}} \tag{60}$$

$$2\hat{\mathbf{w}}^\intercal \mathbf{H}\hat{\mathbf{w}} \geq 2\mathbf{v}^\intercal \mathbf{H}\hat{\mathbf{w}} \tag{61}$$

$$\hat{\mathbf{w}}^\intercal \mathbf{H}\hat{\mathbf{w}} \geq \mathbf{v}^\intercal \mathbf{H}\hat{\mathbf{w}}, \tag{62}$$

where the fourth line uses that $\hat{\mathbf{w}}^\intercal \mathbf{H}\hat{\mathbf{w}} \geq \mathbf{v}^\intercal \mathbf{H}\mathbf{v}$. Therefore

$$(\mathbf{H}\hat{\mathbf{w}})_* = \underset{\|\mathbf{v}\|=1}{\arg\max} \; \mathbf{v}^\intercal \mathbf{H}\hat{\mathbf{w}} = \hat{\mathbf{w}}. \tag{63}$$

$\square$

**Theorem 5.2.** Let $\mathcal{L}(\mathbf{w}) := \frac{1}{2}\mathbf{w}^\top \mathbf{H}\mathbf{w}$ for some $\mathbf{H} \succ 0$. For some norm $\|\cdot\|$, define the generalized sharpness $S := \max_{\|\mathbf{d}\|\leq 1} \mathbf{d}^\top \mathbf{H}\mathbf{d}$. If we run non-Euclidean GD (Def. 1.1) on $\mathcal{L}$, there exists an initialization $\mathbf{w}_0$ from which GD will diverge for any step-size $\eta > 2/S$.

*Proof.* Let $\mathbf{w}_0 \in \text{span}(\hat{\mathbf{d}})$ for some $\hat{\mathbf{d}} \in \underset{\|\mathbf{d}\|=1}{\arg\max} \; \mathbf{d}^\top \mathbf{H}\mathbf{d}$, so $\hat{\mathbf{d}} = \mathbf{w}_0/\|\mathbf{w}_0\|$. We will show $\mathbf{w}_t = (1 - \eta S)^t \mathbf{w}_0$ by induction on $t$. With the property of $\hat{\mathbf{d}}$ from Lemma 5.3, the proof is essentially a direct calculation. From the definition of gradient descent,

$$\mathbf{w}_{t+1} = \mathbf{w}_t - \eta \|\mathbf{H}\mathbf{w}_t\|_* (\mathbf{H}\mathbf{w}_t)_* \tag{64}$$

$$= \|\mathbf{w}_0\|(1 - \eta S)^t \hat{\mathbf{d}} - \eta\|\mathbf{w}_0\|(1 - \eta S)^t \left\|\mathbf{H}\hat{\mathbf{d}}\right\|_* (\|\mathbf{w}_0\|(1 - \eta S)^t \mathbf{H}\hat{\mathbf{d}})_* \tag{65}$$

$$= \|\mathbf{w}_0\|(1 - \eta S)^t \hat{\mathbf{d}} - \eta\|\mathbf{w}_0\|(1 - \eta S)^t \left\|\mathbf{H}\hat{\mathbf{d}}\right\|_* (\mathbf{H}\hat{\mathbf{d}})_* \tag{66}$$

$$= \|\mathbf{w}_0\|(1 - \eta S)^t \hat{\mathbf{d}} - \eta\|\mathbf{w}_0\|(1 - \eta S)^t \left\|\mathbf{H}\hat{\mathbf{d}}\right\|_* \hat{\mathbf{d}} \tag{67}$$

$$= \|\mathbf{w}_0\|(1 - \eta S)^t \left(1 - \eta\|\mathbf{H}\hat{\mathbf{d}}\|_*\right) \hat{\mathbf{d}} \tag{68}$$

$$= \|\mathbf{w}_0\|(1 - \eta S)^{t+1}\hat{\mathbf{d}} \tag{69}$$

$$= (1 - \eta S)^{t+1}\mathbf{w}_0. \tag{70}$$

where the second line uses the inductive hypothesis, the third line uses that the dual map $v \mapsto (v)_*$ is invariant to positive scaling of the input, uses Lemma 5.3, and the fifth line uses

$$\|\mathbf{H}\hat{\mathbf{d}}\|_* = \sup_{\|\mathbf{v}\|=1} \mathbf{v}^\intercal \mathbf{H}\hat{\mathbf{d}} = (\mathbf{H}\hat{\mathbf{d}})_*^\intercal \mathbf{H}\hat{\mathbf{d}} = \hat{\mathbf{d}}^\intercal \mathbf{H}\hat{\mathbf{d}} = \sup_{\|\mathbf{v}\|=1} \mathbf{v}^\intercal \mathbf{H}\mathbf{v} = S. \tag{71}$$

$\square$

As an aside, we can also show that GD will diverge for *every* initialization when $\eta$ is sufficiently large.

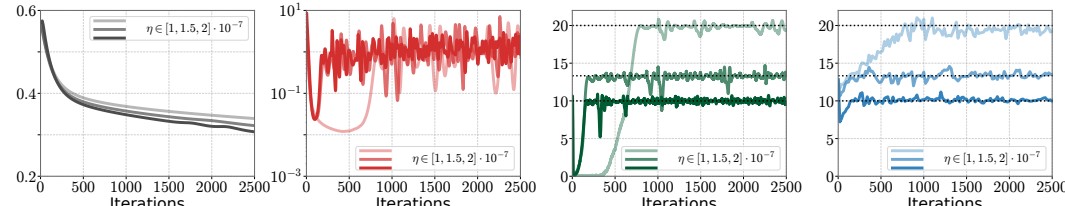

Figure F.1: ($\ell_\infty$-descent) Train loss, gradient norm, directional smoothness, and generalized sharpness (14) during training CNN on CIFAR10-5k with $\ell_\infty$-descent. Horizontal dashed lines correspond to the value $2/\eta$. Gradient norm and train loss curves are smoothed using an exponential smoothing with $\alpha = 0.1$. We use FW with $K = 50$ and $M = 5$ to approximate (14).

**Theorem E.5.** Let $\mathcal{L}(\mathbf{w}) \coloneqq \frac{1}{2}\mathbf{w}^\top \boldsymbol{H}\mathbf{w}$ for some $\boldsymbol{H} \succ 0$. For some norm $\|\cdot\|$, define the generalized sharpness $S^{\|\cdot\|} \coloneqq \max_{\|\mathbf{d}\|\leq 1} \mathbf{d}^\top \boldsymbol{H}\mathbf{d}$. Then, if we run non-Euclidean GD (Definition 1.1) on $\mathcal{L}$, there GD will diverge for every initial point $\mathbf{w}_0$ any step-size $\eta > 2/\mu$.

*Proof.* Starting from the definition of gradient descent,

$$\mathbf{w}_{t+1} = \mathbf{w}_t - \eta\|\boldsymbol{H}\mathbf{w}_t\|_*(\boldsymbol{H}\mathbf{w}_t)_* \tag{72}$$

$$\boldsymbol{H}\mathbf{w}_{t+1} = \boldsymbol{H}\mathbf{w}_t - \eta\|\boldsymbol{H}\mathbf{w}_t\|_*\boldsymbol{H}(\boldsymbol{H}\mathbf{w}_t)_* \tag{73}$$

$$\|\boldsymbol{H}\mathbf{w}_{t+1}\|_* = \left\|\boldsymbol{H}\mathbf{w}_t - \eta\|\boldsymbol{H}\mathbf{w}_t\|_*\boldsymbol{H}(\boldsymbol{H}\mathbf{w}_t)_*\right\|_* \tag{74}$$

$$\|\boldsymbol{H}\mathbf{w}_{t+1}\|_* \geq \eta\|\boldsymbol{H}\mathbf{w}_t\|_*\left\|\boldsymbol{H}(\boldsymbol{H}\mathbf{w}_t)_*\right\|_* - \|\boldsymbol{H}\mathbf{w}_t\|_* \tag{75}$$

$$\|\boldsymbol{H}\mathbf{w}_{t+1}\|_* \geq \left(\eta\left\|\boldsymbol{H}(\boldsymbol{H}\mathbf{w}_t)_*\right\|_* - 1\right)\|\boldsymbol{H}\mathbf{w}_t\|_*. \tag{76}$$

We can bound the coefficient of $\eta$ as

$$\left\|\boldsymbol{H}(\boldsymbol{H}\mathbf{w}_t)_*\right\|_* \geq \inf_{\|\mathbf{v}\|=1}\|\boldsymbol{H}\mathbf{v}\|_* = \inf_{\|\mathbf{v}\|=1}\sup_{\|\mathbf{u}\|=1}\mathbf{u}^\top\boldsymbol{H}\mathbf{v} \geq \inf_{\|\mathbf{v}\|=1}\mathbf{v}^\top\boldsymbol{H}\mathbf{v} = \mu, \tag{77}$$

so

$$\|\boldsymbol{H}\mathbf{w}_{t+1}\|_* \geq (\eta\mu - 1)\|\boldsymbol{H}\mathbf{w}_t\|_*, \tag{78}$$

and therefore

$$\|\boldsymbol{H}\mathbf{w}_t\|_* \geq (\eta\mu - 1)^t\|\boldsymbol{H}\mathbf{w}_0\|_*. \tag{79}$$

Since $\eta > 2/\mu \implies \eta\mu - 1 > 1$, the parameter norm $\|\boldsymbol{H}\mathbf{w}_t\|_*$ increases exponentially, and GD diverges. $\qquad\square$

# F ADDITIONAL EXPERIMENTAL RESULTS WITH $\ell_\infty$ DESCENT

## F.1 CONVERGENCE WHEN TRAINING CNN MODEL

## F.2 SENSITIVITY OF FRANK-WOLFE ALGORITHM IN ESTIMATING THE GENERALIZED SHARPNESS FOR SIGN GRADIENT DESCENT

In this section, we study the sensitivity of the Frank-Wolfe algorithm in estimating the generalized sharpness of non-Euclidean gradient descent methods. Our experiments are conducted on a CNN with two convolutional layers, followed by a linear layer, trained on the CIFAR10-5k dataset (Krizhevsky & Hinton, 2009). We run $\ell_\infty$-descent, and approximate the generalized sharpness by Frank-Wolfe with 50 iterations, using $\{1, 7, 15\}$ initialization points drawn from a standard normal distribution, and take the maximum over restarts as the generalized sharpness estimate.

In Fig. F.2, we show that the Frank-Wolfe estimate of the generalized sharpness is sensitive to the number of restarts. With a single random initialization, the algorithm generally underestimates the

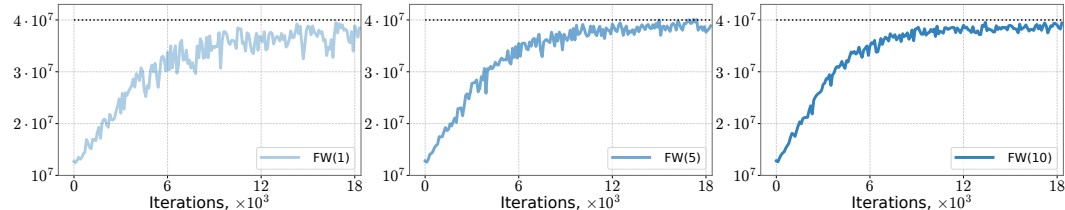

Figure F.2: The approximation of the generalized sharpness of $\ell_\infty$-descent by the Frank-Wolfe algorithm varying the number of initialization points in $\{1, 5, 10\}$ for the Frank-Wolfe algorithm. Here, FW($k$) denotes $k$ restarts of the Frank-Wolfe algorithm, with varying initialization points.

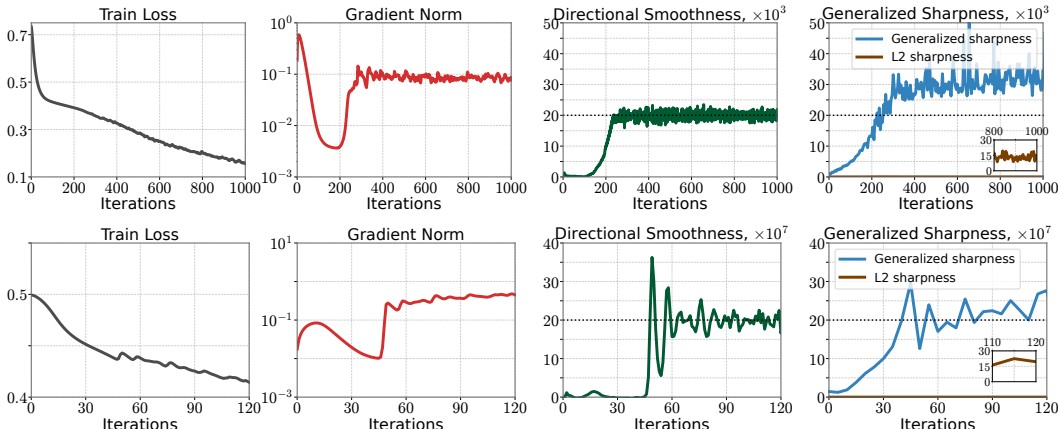

Figure F.3: ($\ell_\infty$-descent) Train loss, gradient norm, directional smoothness, generalized sharpness (14), and L2 sharpness ($\lambda_{\max}(\nabla^2 \mathcal{L}(\mathbf{w}_t))$) during training Resnet20 (top, $\eta = 10^{-4}$) and VGG11 (bottom, $\eta = 10^{-7}$) on CIFAR10 with $\ell_\infty$-descent. Horizontal dashed lines correspond to the value $2/\eta$.

value. Increasing the number of restarts to $15$ yields a much more stable estimate that closely aligns with the true value almost everywhere.

### F.3 RESULTS ON RESNET20 AND VGG11

In this section, we provide additional empirical results on larger models, such as Resnet20 (He et al., 2016) and VGG11 (Simonyan & Zisserman, 2014), trained on the CIFAR10 dataset with $\ell_\infty$-descent and MSE loss. From the results in Figure H.5, we observe that both directional smoothness and generalized sharpness hover at the stability threshold $2/\eta$. In contrast, a standard notion of sharpness, i.e., $\lambda_{\max}(\nabla^2 \mathcal{L}(\mathbf{w}_t))$ defined in the Euclidean norm, lies significantly below the threshold (brown line in the right subfigure). Note that for Resnet20 model, the generalized sharpness stabilizes slightly above the threshold due to several unstable directions as explained in Section C.

## G ADDITIONAL EXPERIMENTAL RESULTS WITH BLOCK GRADIENT DESCENT

### G.1 TRAINING DETAILS

Our implementation is based on open source code from Cohen et al. (2021) together with publicly available datasets. In all our experiments, we use algorithms with full-batch gradient, i.e., we run them in the deterministic setting. The datasets and step-sizes $\eta$ used in the experiments are specified in the figures. In not specified, we use the Frank-Wolfe algrorithm with $M = 5$ restarts and $K = 50$ iterations, and `PolarExpress` with 5 steps.

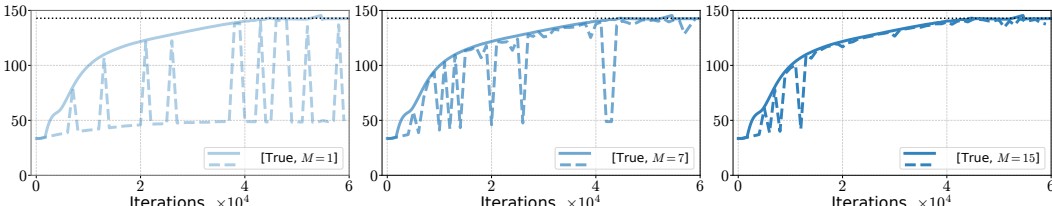

Figure G.1: The maximum block-wise Hessian eigenvalue (solid line), which is the generalized sharpness of `Block CD`, and its approximation by the Frank-Wolfe algorithm varying the number of initialization points in $\{1, 7, 15\}$ for the Frank-Wolfe algorithm. Here, $M$ is the number of restarts of the Frank-Wolfe algorithm, varying the initialization point.

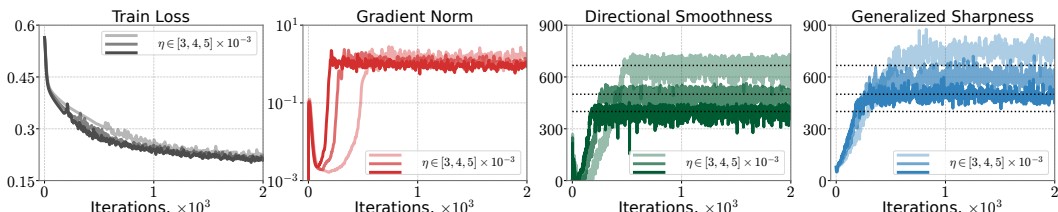

Figure H.1: (`Spectral GD`) Train loss, gradient norm, directional smoothness, and generalized sharpness (19) during training CNN model on CIFAR10 dataset with the `Spectral GD`. Horizontal dashed lines correspond to the value $2/\eta$.

In the training of CNN and MLP models, we use MSE loss, while in the training of the Transformer model, we use a rescaled MSE loss from Hui & Belkin (2020).

## G.2 SENSITIVITY OF FRANK-WOLFE ALGORITHM IN ESTIMATING THE GENERALIZED SHARPNESS FOR BLOCK GRADIENT DESCENT

In this section, we study the sensitivity of the Frank-Wolfe algorithm in estimating the generalized sharpness of non-Euclidean gradient descent methods. Our experiments are conducted on a CNN with four convolutional layers, followed by a linear layer, trained on the CIFAR10-5k dataset (Krizhevsky & Hinton, 2009). Now we evaluate Block GD, where the generalized sharpness has a closed-form expression (30). We run Frank-Wolfe for 50 iterations, using $\{1, 7, 15\}$ initialization points drawn from a standard normal distribution, and take the maximum over restarts as the generalized sharpness estimate. The Frank-Wolfe procedure is applied every 100 iterations of `Block CD`.

In Fig. G.1, we show that the Frank-Wolfe estimate of the maximum block-wise Hessian eigenvalue is sensitive to the number of restarts. With a single random initialization, the algorithm provides a good approximation at a few iterations but generally underestimates the value. Increasing the number of restarts to 15 yields a much more stable estimate that closely aligns with the true value almost everywhere.

# H ADDITIONAL EXPERIMENTAL RESULTS WITH SPECTRAL GRADIENT DESCENT

## H.1 CONVERGENCE WHEN TRAINING CNN MODEL

In this section, we present the results when training CNN model on CIFAR10 dataset with `Spectral GD`; see Figure H.1. The results support our theoretical observations.

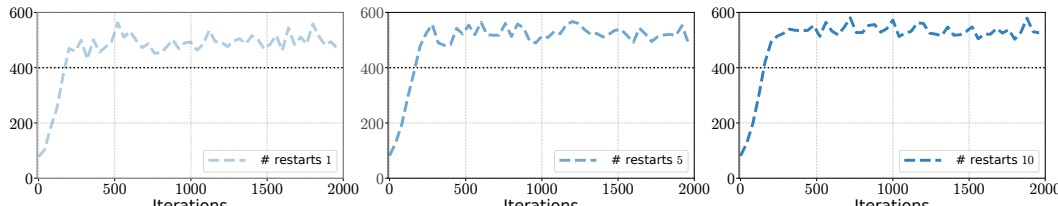

Figure H.2: The approximation of the generalized sharpness by the Frank-Wolfe algorithm for `Spectral GD` varying the number of initialization points in $\{1, 5, 10\}$ for the Frank-Wolfe algorithm.

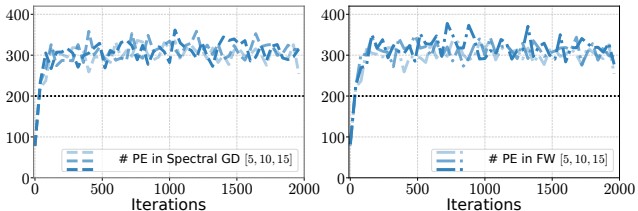

Figure H.3: The sensitivity of the generalized sharpness estimation of `Spectral GD` to the number of Polar Express steps in `Spectral GD` (left) and in Frank-Wolfe (right). Here # PE means the number of Polar Express steps in `Spectral GD` or Frank-Wolfe algorithm respectively.

### H.2 SENSITIVITY OF FRANK-WOLFE ALGORITHM IN ESTIMATING THE GENERALIZED SHARPNESS FOR SPECTRAL GRADIENT DESCENT

Next, we switch to the `Spectral GD` to train CNN model on the full CIFAR10 dataset. We perform a similar procedure to the one done in the previous section. We fix the number of Polar Express steps in both `Spectral GD` and Frank-Wolfe to 5 and vary the number of initialization points for Frank-Wolfe in $\{1, 5, 10\}$. Each run of Frank-Wolfe has 50 iterations.

In Fig. H.2, we observe that `Spectral GD` is less sensitive to the number of initialization points for Frank-Wolfe than Block GD. Therefore, it is not necessary to do restarts for Frank-Wolfe when it is used to measure the generalized sharpness of the `Spectral GD` algorithm.

### H.3 SENSITIVITY OF SPECTRAL GRADIENT DESCENT TO THE NUMBER OF POLAR EXPRESS STEPS

We investigate how the number of Polar Express steps affects the generalized sharpness estimation of `Spectral GD`. To this end, we fix the number of Polar Express steps in `Spectral GD` and vary the number of steps in the Frank-Wolfe algorithm across $\{5, 10, 15\}$, and vice versa. All experiments are conducted using a CNN with four convolutional layers, trained on the full CIFAR-10 dataset.

As shown in Fig. H.3, we do not observe any significant differences across the different configurations. This indicates that 5 steps of the Polar Express algorithm are sufficient to obtain an accurate and stable estimate of `Spectral GD`'s generalized sharpness.

### H.4 QUADRATIC TAYLOR APPROXIMATION OF THE REAL OBJECTIVE

### H.5 RESULTS ON RESNET20 AND VGG11

In this section, we provide additional empirical results on larger models, including ResNet20 (He et al., 2016) and VGG11 (Simonyan & Zisserman, 2014), trained on the CIFAR10 dataset using `Spectral GD` with MSE loss. As shown in Figure H.5, both the directional smoothness and the generalized sharpness remain close to the stability threshold $2/\eta$. In contrast, the standard notion of sharpness—namely $\lambda_{\max}(\mathcal{L}(\mathbf{w}_t))$ computed in the Euclidean norm—stays well below this thresh-

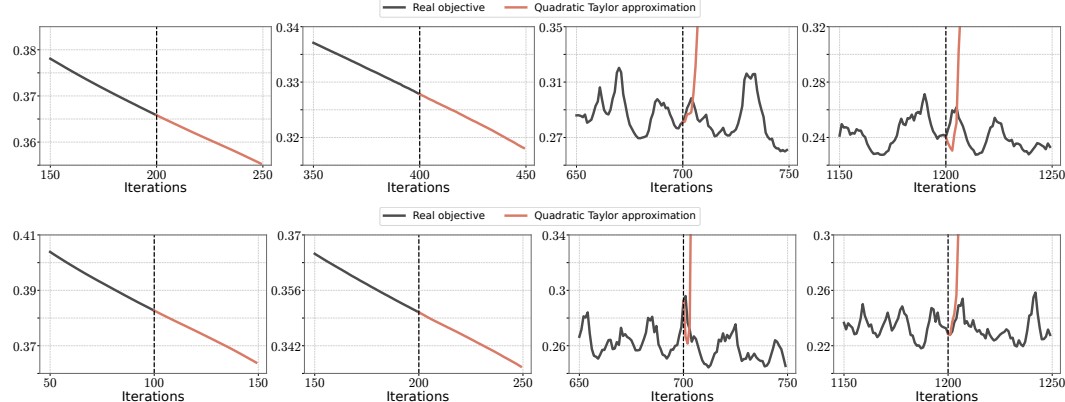

Figure H.4: MSE loss (top row $\eta = 0.003$, bottom row $\eta = 0.004$). At 4 different iterations during the training of the CNN from Fig. 4 (marked by the vertical dotted black lines), we switch from running `Spectral GD` on the real neural training objective (for which the train loss is plotted in gray) to running `Spectral GD` on the quadratic Taylor approximation around the current iterate (for which the train loss is plotted in orange). Two left figures are timesteps before `Spectral GD` has entered EoS; observe that the orange line (Taylor approximation) closely tracks the blue line (real objective). Two right figures are timesteps during the EoS; observe that the orange line quickly diverges, whereas the blue line does not.

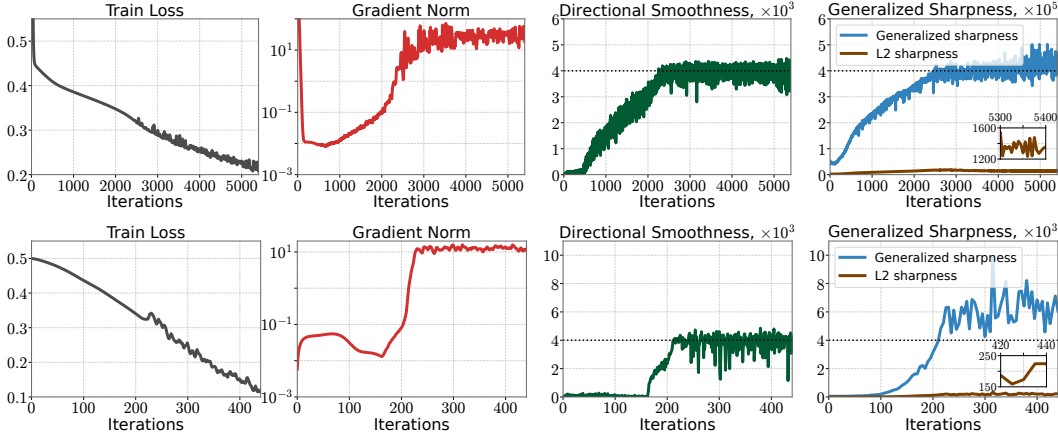

Figure H.5: (`Spectral GD`) Train loss, gradient norm, directional smoothness, generalized sharpness (14), and L2 sharpness ($\lambda_{\max}(\nabla^2 \mathcal{L}(\mathbf{w}_t))$) during training Resnet20 (top, $\eta = 5 \cdot 10^{-5}$) and VGG11 (bottom, $\eta = 5 \cdot 10^{-4}$) on CIFAR10 with $\ell_\infty$-descent. Horizontal dashed lines correspond to the value $2/\eta$.

old (brown curve in the right panel). For the ResNet20 model, the generalized sharpness stabilizes slightly above $2/\eta$, which can be attributed to the presence of several unstable directions, as discussed in Section C.

## I $\quad \ell_\infty$-DESCENT AND RMSPROP

In this section, we report results for the `RMSprop` algorithm when training an MLP on the CIFAR10-5k subset with MSE loss. Although `SignGD` can be viewed as a limiting case of `RMSprop` as $\beta_2 \to 0$, the adaptive EoS (AEoS) condition of Cohen et al. (2022) is valid only when $\beta_2$ is large (i.e., close to 1 in practical settings) and breaks down as $\beta_2$ becomes small. For small $\beta_2$, the largest eigenvalue of the preconditioned Hessian $\lambda_{\max}(P_t^{-1}\nabla^2 \mathcal{L}(\mathbf{w}_t))$ does not stabilize around $2/\eta$; instead, it often exceeds this value by a substantial margin. The underlying issue is that as $\beta_2 \to 0$, the algorithm no

longer resembles preconditioned gradient descent with a slowly-changing preconditioner, which is the approximation that inspires the `AEoS` condition.

Our results in Figure I.1 support this observation. We plot the top four eigenvalues of the preconditioned Hessian for `RMSprop`, showing that they stabilize around the threshold $2/\eta$ only when $\beta_2$ is large, while for small $\beta_2$ the behavior deviates significantly.

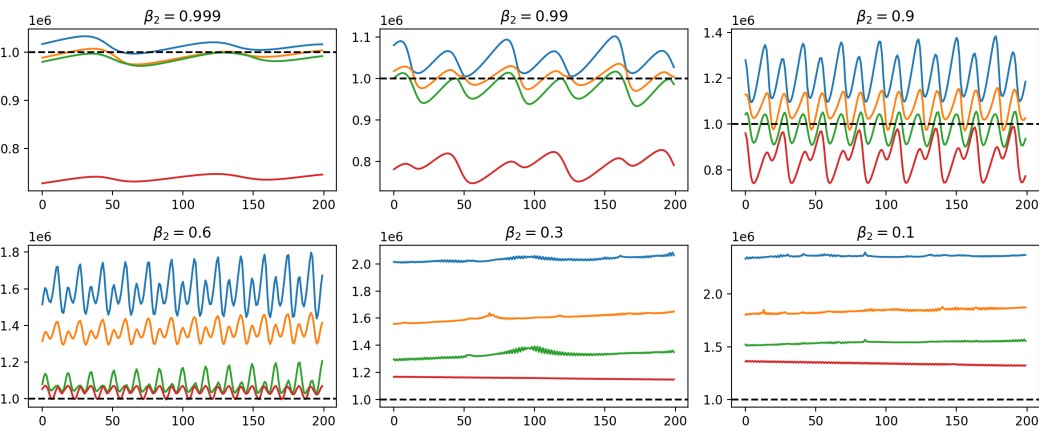

Figure I.1: Sharpness of `RMSprop` when training MLP model on a subset of CIFAR10 dataset, varying $\beta_2$ hyperparameter. Here, colored lines correspond to the evolution of the top-4 largest eigenvalues of the preconditioned Hessian, while the dashed line is $2/\eta$ threshold. We observe that `RMSprop` reaches `AEoS` only for realistic (close to 1) values of $\beta_2$, while for small $\beta_2$ the preconditioned sharpness is not at $2/\eta$, but significantly higher.

