# OpenReview forum: "Non-Euclidian Gradient Descent Occurs at the Edge of Stability"
_ICLR.cc/2026/Conference — Submitted to ICLR 2026_

### Official Review · Reviewer_ybaT · 2025-10-28

**Soundness:** 2
**Presentation:** 3
**Contribution:** 2
**Rating:** 4
**Confidence:** 4

**Summary:**

The paper extends the *Edge of Stability* (EoS) phenomenon to non-Euclidean gradient descent. It defines *sharpness* under general norms via directional smoothness and shows that several non-Euclidean optimizers (ℓ∞, spectral, block ℓ1,2) also exhibit EoS behavior. A simple theoretical analysis on convex quadratics demonstrates convergence when sharpness < 2/η and divergence otherwise.

**Strengths:**

1. Defines a generalized notion of sharpness under arbitrary norms, conceptually clear and well-motivated.
2. Demonstrates EoS in various non-Euclidean settings, suggesting generality beyond Euclidean GD.

**Weaknesses:**

1. Experiments too simple – limited to small models and datasets, lacking the breadth of prior EoS studies (e.g., Cohen et al.).
2. Theory too shallow – results proven only for convex quadratics; lacks generalization to non-convex deep learning settings.
3. Insufficient depth – does not explain the mechanism behind non-Euclidean EoS or analyze differences among norms.

**Questions:**

Is it possible to prove the theory in a setting that is closer to real neural networks, rather than on simple convex functions?

Also see Weakness.

---

> ### Author Response · Authors · 2025-11-21
> **Rebuttals (part 1)**
>
> We thank the reviewer for the thoughtful and encouraging comments. We appreciate the recognition of the clarity of our paper and the acknowledgment of the strength of our motivation and the generality of our approach.
>
> We hope that the explanations provided below will assist in re-evaluating our submission and possibly revising the score.
>
> **Q1: Experiments are too simple.**
>
>  ***A1:*** We emphasize that our empirical study covers three types of architectures (CNN, MLP, and Transformer), three distinct non-Euclidean descent algorithms, including their normalized variants, and two datasets, namely CIFAR-10 and Tiny Shakespeare. We also note that in the experiments with Spectral Descent, we used the full CIFAR-10 dataset containing 50,000 samples and the Tiny Shakespeare dataset containing one million samples. We acknowledge that our experiments are restricted to relatively small models because computing full Hessian vector products is computationally demanding, but they are of the same scale as in [Cohen et al., 2021]. Going beyond small-scale models would require significantly better computing resources. In addition, we include ablation studies that present results on the sensitivity of the Frank–Wolfe algorithm to the choice of norm and hyperparameters, experiments based on quadratic approximations of the loss function, and comparisons between the $\ell_{\infty}$ and $\ell_2$ norms (see Section B). Finally, we add additional results in section I that shed light on a clean distinction between RMSprop when $\beta_2\to0$ and SignGD. Therefore, we believe that the empirical results we provide are sufficient to support our theoretical findings across different norms, datasets, and model architectures.
>
> However, if the reviewer has some specific experimental setup and why it should have been tested, we are happy to discuss!
>
>
> **Q2: Theory is too shallow**
>
> ***A2:*** For clarity, we break down our answer below.
>
>  - **a.** Our analysis of directional smoothness and its connection to sharpness is valid for any twice continuously differentiable function, which includes most large language models.
>
> - **b.** A rigorous convergence analysis of the EoS phenomenon in deep learning is still lacking, even in the Euclidean setting. In most cases, existing theory either focuses on quadratic objectives, as we do, or relies on simplified models (e.g., central flow [Cohen et al., 2024]) that do not provide strong theoretical guarantees. See also our response to ***A4***.
>
> - **c.** Furthermore, a comprehensive theoretical understanding of gradient descent in the non-convex deep learning regime remains essentially an open problem. By meaningful analysis, we refer to theoretical results based on assumptions that reflect empirically observed behaviors in deep learning. Although there exist analyses in the general smooth non-convex setting, they typically rely on unrealistic assumptions and provide overly pessimistic bounds that do not mirror well empirical behaviour.
>
> **Q3: Insufficient depth.**
>
> ***A3:*** We do shed light on the underlying mechanism: if the loss exhibits oscillations, then the directional smoothness must oscillate around $2/\eta$, and consequently, the generalized sharpness oscillates just above $2/\eta$. We further illustrate how this behavior manifests under different norms and how the generalized sharpness varies accordingly. This explanation holds for any twice continuously differentiable function, which includes most neural networks. Additionally, we provide empirical evidence highlighting the differences between norms (see Section B), and examine the sensitivity (or its absence) of the generalized sharpness estimation procedure using the Frank–Wolfe algorithm across different norms. In Section I, we included the results that shed light on the differences between RMSprop and SignGD, highlighting that prior studies do not cover the algorithms analyzed in our work and underscoring the significance of our contributions.
>
> The exact reason for the oscillations observed at the EoS remains an open question. Nevertheless, it is worth noting that even in the Euclidean setting, this phenomenon was not explained in early studies and required time to develop a proper theoretical understanding. Therefore, we leave such an explanation to future work.

---

> > ### Author Response · Authors · 2025-11-21
> > **Rebuttals (part 2)**
> >
> > **Q4: Theory in more realistic setting.**
> >
> > ***A4:*** Our main insights, and connections between the behaviour of the loss and (generalized) sharpness hold for all twice continuously differentiable functions.
> >
> > We acknowledge that extending the theoretical results to a broader class of functions would further strengthen our work. However, analyzing the convergence/divergence behavior of non-Euclidean descent methods is already a challenging problem even in the quadratic case, and it is an interesting research direction in its own right. We emphasize that the for arbitrary norm, the analysis is significantly more difficult than that of Euclidean descent, even for quadratic objectives, since the spectral quantities of the Hessian in this case are less studied (e.g., what are the generalizations of eigenvectors? do they form an orthogonal basis?; see our responses ***A2*** and ***A10*** for reviewer QE9g for a more detailed explanation). Nevertheless, our theoretical results on quadratics align well with our empirical findings, particularly in supporting our choice of the generalized sharpness measure. Moreover, we emphasize that quadratic functions serve as a standard and insightful model for studying the EoS phenomenon, since any smooth function is locally quadratic. They provide an intuitive framework for understanding EoS behavior (see Figures 6 and H.4). More complex models considered in the literature (e.g., Cohen et al., 2024, Understanding Optimization in Deep Learning with Central Flow) are primarily used to provide more insights about the EoS phenomenon rather than formal theoretical guarantees.

---

> > > ### Comment · Reviewer_ybaT · 2025-11-24
> > >
> > > Thank you for the detailed response. Below is my explanation of my concerns and suggestions.
> > >
> > > First, I would like to clarify that my comment about the theory being insufficient was not questioning the definition of sharpness on twice continuously differentiable functions. My concern is that the convergence analysis is limited to convex quadratic optimization. In this setting, with a fixed learning rate, the dynamics essentially reduce to either divergence or convergence, and an edge-of-stability phenomenon is unlikely to appear. This creates a fundamental gap between the analysis provided and the actual EoS behavior observed in practice.
> > >
> > > In addition, although the authors argue that their experimental scale is comparable to Cohen et al. (2021), I do not share this view. Cohen et al. validated their findings on practical VGG and ResNet models, whereas this submission only evaluates a very small CNN. Their transformer experiments were conducted on the commonly used Wikitext-2 dataset, while the present work relies on Tiny Shakespeare, which is far smaller and less representative. Moreover, the experimental breadth in Cohen et al. is significantly larger:
> > > 1. They tested across multiple CIFAR-10 subsets and also on polynomial-fitting tasks, demonstrating dataset-level universality of EoS.
> > > 2. They varied model architectures (depth, width, activation functions) to show robustness across model choices.
> > > 3. They explored a wide range of momentum algorithms, verifying the generality of EoS with respect to optimization methods.
> > >
> > > From a contribution standpoint, Cohen et al. (2021) were the first to discover and characterize EoS, whereas this paper mainly extends EoS observations to different norms. While insightful, this extension alone is somewhat limited in contribution.
> > >
> > > If the authors are willing to include the following experiments, I would be happy to raise my score:
> > >
> > > 1. ResNet and VGG on CIFAR-10.
> > >
> > > 2. A transformer model on Wikitext-2.
> > >
> > > 3. For the small-scale experiments in the paper (MLP and the small CNN), it would be important to also measure the trajectory-based L2 sharpness (i.e., standard EoS sharpness). As an extension of EoS-related work, this is necessary. Furthermore, comparing the L2 sharpness of these new algorithms with that of GD could help reveal how the implicit bias of methods such as SignGD and Spectral GD differs from that of GD, which would significantly strengthen the contribution of the paper.

---

> ### Author Response · Authors · 2025-12-03
> **Additional results**
>
> Following the reviewer's request, we attached the results for $\ell_\{\infty}$-descent and Spectral GD when training Resnet20 and VGG11 models on the CIFAR10 dataset, supporting our claims beyond small-scale MLP and CNN models. The results are presented in Section F.3 and Section H.5, respectively. We refer the reviewer to Figures F.3 and H.5 for the empirical results. We highlight that our claims are still applicable in these settings as well. Note that in the right plots of Figures F.3 and H.5 we also demonstrate the dynamics of $\ell_2$-sharpness (defined in the Euclidean norm) for comparison as the reviewer requested. We observe that it lies significantly below the stability threshold $2/\eta$ while the proposed generalized sharpness hovers at $2/\eta$ or slightly above, which is dictated by several unstable directions (explained in Section C).
>
> We emphasize that all the changes are highlighted in blue for clarity.
>
> We hope that these additional experimental results will assist in re-evaluating our submission and possibly revising the score. Unfortunately, we are not able to provide the results on Wikitext-2 by the deadline due to time and resource constraints.

---

### Official Review · Reviewer_QE9g · 2025-11-01

**Soundness:** 3
**Presentation:** 3
**Contribution:** 3
**Rating:** 6
**Confidence:** 5

**Summary:**

The authors investigate whether a wider class of gradient based methods typically occurr at the EoS.
They show that they do.

**Strengths:**

The work is very interesting. The message is clear, clean, and nice (maybe you could put (10) in page 1 so that one gets the message straight away). I really liked the paper.
I also think (as a person working on the topic) this is something someone had to do! So good job!

I think there are a few weaknesses which necessitate a little further work. If the authors will be able to pull this work, the paper will be great and I will gladly support the acceptance!

**Weaknesses:**

### **On the quantities being certificates and not causes**

In searching for a quantity that gets at the edge of stability, if we want to construct a theory in the fashion of self-stabilization of Damian et al. (2023) or Central Flow of Cohen et al. (2024), I believe we want a curvature quantity which does not depend on the weights oscillating, it's just a curvature feature of the points along the central flow. I don't know if I am explaining myself correctly.
For instance, for GD, $\lambda_{\max}$ is not a property of the gradients or the weights, it is a property of the Hessian in the point in the middle of the oscillations. Importantly your quantities depend on a direction too, this is a directional curvature as Lee and Jang (2023). If we believe the self-stabilization/central flow story the curvature quantity needs to be a value independent of the gradient/weights directions. This is not the case for these quantities, so I wonder if these are just *certificates* of EoS happening but are not *causal*.

*Meaning these geometrical properties may be consequences of EoS, not EoS being a consequence of these geometrical properties.*

Can you please comment on this? Happy to develop further.

In the case of Andreyev and Beneventano (2024) for SGD I am convinced that the *"real, causal,"* quantity, if any, could be an expectation of a Rayleigh quotient, because in the mini-batch case the gradients *will not* align along a precise direction. But for deterministic methods, I expect the gradients/weights to align with the eigenvector of a certain quantity, I want that quantity which is independent of alignment (as $\lambda_{\max}$), not a quantity subject to alignment as a Rayleigh quotient.

Does this make sense? Is there a reason why you think that my point is wrong?

Imagine I have a quadratic, one eigenvalue is bigger than $2/\eta$ and the other is smaller. I expect Muon to adapt the gradients in a way *due to* that eigenvalue, in that case when I develop a self-stabilization theory for Muon I would use that eigenvalue, not the Rayleigh quotient. Right? Or do you think it is not self-stabilization the machinery that induces EoS?

Thanks a lot in advance for considering to clarify, I'm genuinely interested in what you think!

### **The proofs in the appendix**

I feel like those proofs in the appendix are unnecessary, they either exist already or are not needed in this article. Just cut them, it will be better and shorter.

### **More experiments?**

Even more in the light of the comment above on certificate vs cause, I believe I would like you to show more experiments both on quadratics, both on Cifar10 or other dataset. I believe you should go bigger both in the models and in the datasets to have this result properly established! Papers as Cohen et al. (2021), Andreyev and Beneventano (2024), Cohen et al (2024) have several dozen of pages of Appendix in which they show different things. I believe it would be beneficial to do the same. For instance, checking also how the trace goes up or other eigenvalues, etc, would help having a cleaner picture. My acceptance grade will depend on this too.

**Questions:**

See weaknesses too :)

1 ) The directional smoothness sounds a lot like the Rayleigh quotient of gradients and Hessian. The fact that that Rayleigh quotient stabilizes better than $\lambda_{\max}(H)$ for GD was already established by Lee and Jang (2023) and possibly by Cohen et al. (2021). Can you please shed light on their relation? I feel like this thing of the generalized shaprness seen as a Rayleigh quotient was known for GD, am I wrong?

2) In case, even the quantity of Andreyev and Beneventano (2024) is a form of Rayleigh quotient, did you consider running experiments on the  expected mini-batch landscape as they did?

3) Cohen et al. (2022) computes AEoS for Adam. Adam in practice is meant to work approximately similarly to a sign descent. You however, notice that sign descent work differently. Do you have any idea to bridge these two results? It would be interesting!

4) Can you give me some better intuition of the sudden drops of which you speak in lines 318-320?

5) I believe the preoscillatory regime is standard, it should be the case also of GD, am I wrong? For instance if $\eta = 1.5/\lambda_{\max}$. Or at least, in parameter space there should be oscillation as you cross $1/\lambda_{\max}$, am I wrong? I am extremely curious of what your intuition is for this phase if you think it is not that!
PS: At line 772 you start without capitalizing.

6) It would be intersting to see if something like what happen in GD with the second eigenvalue happens. What is the second quantity that goes up? What the third? Is there a nice characterization of this? I believe there is, happy to throw my idea at you during rebuttal!

---

> ### Author Response · Authors · 2025-11-21
> **Rebuttals (part 1)**
>
> We thank the reviewer for the thoughtful and encouraging comments. We appreciate the recognition of the clarity of our paper and the acknowledgment that it addresses an important and interesting topic.
>
> We hope that the explanations provided below will assist in re-evaluating our submission and possibly revising the score.
>
> **Q1: geometrical properties may be consequences of EoS, not EoS being a consequence of these geometrical properties.**
>
>  ***A1:*** Thank you for raising this point. We agree with the critique to some extent. Directional smoothness is not the appropriate property in the strict sense. However, we use directional smoothness only as an initial stepping stone to motivate what we believe is a suitable candidate for generalized smoothness. Our generalized smoothness definition in equation (10) is independent of the gradient or the update direction and depends only on the spectral characteristics of the loss function.
>
> **Q2: Further discussion around Andreyev and Beneventano (2024) and quadratics.**
>
> ***A2:*** Thanks so much for the engaging questions and points! As for the stochastic setting, we find that the extension to the non-Euclidean setting was already novel and challenging. Thus, we focus on the deterministic full batch setting. In the deterministic non-Euclidean setting, we found that the gradients/weights tend to align with invariant directions of the update, not eigenvectors of the Hessian. Essentially, we found that the dynamics tend to operate in terms of a non-Euclidean generalization of eigenvalues/eigenvectors, which we do not yet fully understand. Let us elaborate.
>
> Let's look at the simple case of sign descent on a quadratic $F(w) = \frac{1}{2} w^\top Aw$ for a $d \times d$ matrix $A$. The invariant directions $v$ are those for which a single step of sign descent from $v$ will map somewhere parallel to $v$. More formally, $v - \eta \lVert \nabla F(v) \rVert_1 \text{sign}(\nabla F(v))$ is parallel to $v$, or equivalently $\text{sign}(Av) = \lambda v$ for some scalar $\lambda$. You can see how this is a generalization of eigenvectors/eigenvalues in terms of the $\ell_{\infty}$ geometry. We found that for quadratics, the weights/gradients tend to align with these invariant directions, not with eigenvectors of $A$. The issue is that for non-Euclidean geometries, the set of such invariant directions is much harder to characterize than in the Euclidean case. In the Euclidean case, there are always $d$ eigenvectors of $A$ that form an orthogonal basis, so it is quite easy to characterize the dynamics of GD in terms of these invariant directions. In the non-Euclidean case, there may be anywhere from $1$ to $d$ invariant directions, and they are not necessarily orthogonal. Luckily, we can prove that there is at least one invariant direction (this is our Lemma 5.3), which is the maximizer of the Rayleigh quotient and is the generalization of the top eigenvector from the Euclidean case.
>
> So to answer your question about a certain quantity which is independent of alignment (as $\lambda_{\max}$), we think that the maximum value of the Rayleigh quotient (i.e. our definition of generalized sharpness in Definition 2.2) is a good starting point, but obviously it does not satisfy your desired property of depending only on the points along the central flow. It seems in principle possible to develop a self-stabilization theory for Muon, but even before this, we must understand the dynamics on the quadratics. We guess that such a theory should be in terms of these invariant directions (the "non-Euclidean eigenvectors"), and this is beyond the scope of our initial work on identifying an edge of stability phenomenon for non-Euclidean gradient methods.
>
> Again, thanks a lot for the interesting discussion, and please let us know if you have more questions.
>
> **Q3: Proofs are unnecessary.**
>
> **A3:** We highlight that the proofs for convergence of non-Euclidean gradient descent on quadratics (Section E) are an important part of our work, and most of the results of that section are novel. They use generalizations of smoothness and strong convexity to an arbitrary norm, followed by the actual convergence proofs. We acknowledge that the results of Section D are partially known, but we add the proofs for completeness of the presentation and easier reading.

---

> > ### Author Response · Authors · 2025-11-21
> > **Rebuttals (part 2)**
> >
> > **Q4: More empirical evidence is required.**
> >
> > ***A4:***
> > - **a.** We emphasize that our empirical study covers three types of architectures (CNN, MLP, and Transformer), three distinct non-Euclidean descent algorithms, including their normalized variants, and two datasets, namely CIFAR-10 and Tiny Shakespeare. We also note that in the experiments with Spectral Descent, we used the full CIFAR-10 dataset containing 50,000 samples and the Tiny Shakespeare dataset containing one million samples. We acknowledge that our experiments are restricted to relatively small models because computing full Hessian vector products is computationally demanding, but they are of the same scale as in [Cohen et al., 2021]. Going beyond small-scale models would require significantly better computing resources. In addition, we include ablation studies that present results on the sensitivity of the Frank–Wolfe algorithm to the choice of norm and hyperparameters, experiments based on quadratic approximations of the loss function, and comparisons between the $\ell_{\infty}$ and $\ell_2$ norms (see Section B). Therefore, we believe that the empirical results we provide are sufficient to support our theoretical findings across different norms, datasets, and model architectures.
> > - **b.** In Section I, we also included experiments using RMSprop to train the MLP model. These results show that adaptive EoS applies only when $\beta_2\approx 1$ and that there is a clear distinction between the spectral behavior of $\ell\_{\infty}$-descent and RMSprop. This highlights that prior studies do not cover the algorithms analyzed in our work and underscores the significance of our contributions.
> > - **c.** In response ***A2***, we explained that the generalization of eigenvectors and eigenvalues to the non-Euclidean setting is not well understood and requires further investigation (see also our response to ***A10***). Therefore, the suggested experiments lie beyond the scope of this work. We believe that the trace and eigenspectrum of the Hessian, as defined under the Euclidean norm, are not the appropriate quantities to compute in the context of non-Euclidean norms.
> > - **d.** If the reviewer has other suggestions for experiments, we would be happy to consider them.
> >
> > **Q5: Relation between the Rayleigh quotient and directional smoothness.**
> >
> > ***A5:*** There is indeed a strong relation between directional smoothness and the Rayleigh quotient. If $d_t$ is the update direction (e.g., negative gradient in the Euclidean case), then the directional smoothness is $D^{\\|\cdot\\|}(w_t,w_{t+1}) = \frac{\int_0^1d_t^\top \nabla^2\mathcal{L}(w_t-\tau \eta d_t)d_t\mathrm{d}\tau }{\|d_t\|^2}$, while the Rayleigh quotient is $R(w_t) = \frac{d_t^\top \nabla^2\mathcal{L}(w_t)d_t}{\|d_t\|^2}$. Therefore, these two quantities are almost the same assuming that the Hessian does not change much along the segment $[w_t, w_{t+1}]$ and the stepsize $\eta$ is small enough. Finally, the generalized sharpness in (10) is a maximum of the Rayleigh quotient over all possible directions.
> >
> > **Q6: Stochastic experiments.**
> >
> > ***A6:*** As mentioned in A2, before addressing the stochastic setting, it is important to first understand the EoS behavior of non-Euclidean descent methods in the deterministic case, since this problem is already highly challenging. We leave the investigation of the stochastic setting for future work.
> >
> > **Q7: Difference between Adam and SignGD**
> >
> > ***A7:*** This is an interesting question!  While it's true that normalized sign descent is the $\beta_2 = 0$ limit of RMSProp, the AEoS condition from Cohen et al. (2022) only applies at large (realistic) values of $\beta_2$ and breaks down as $\beta_2$ gets close to zero.  In particular, for small $\beta_2$, the largest eigenvalue of $\lambda_{\max}(\text{diag}(\nu_t)^{-1/2} H(w_t))$ is not at $2/\eta$, and is instead rather often far above this value.  The issue is that when $\beta_2$ approaches 0, the algorithm no longer resembles preconditioned gradient descent with a slowly-changing preconditioner, which is the approximation that inspires the AEoS condition. We include these results in Section I of the revision. The changes are highlighted in blue.

---

> > > ### Author Response · Authors · 2025-11-21
> > > **Rebuttals (part 3)**
> > >
> > > **Q8: Explanation of sudden drops with Block GD.**
> > >
> > > ***A8:*** Let us clarify this phenomenon in more detail. Note that for Block GD, we update only one block at each iteration, which has the largest gradient norm. Let the block $\ell_1\in[L]$ reach EoS the first at iteration $t_1$, i.e. $\lambda_{\max}(\nabla^2_{w^{\ell_1}}(w_{t_1})) > 2/\eta$. At this stage, other blocks did not reach EoS yet, i.e. $\lambda_{\max}(\nabla^2_{w^\ell}L(w_{t_1})) < 2/\eta$ for all $\ell\in[L], \ell\neq \ell_1$. We stay at EoS as long as Block GD keeps updating the block $\ell_1$. Let at iteration $t_2$, Block GD chose to update another block $\ell_2\neq\ell_1$. Since the block $\ell_2$ is not at EoS, the directional smoothness drops back close to $0$. At the same time, the generalized sharpness does not go to $0$ because the block $\ell_1$ is still at EoS (its eigenvalue is above $2/\eta$) and its weights are not updated at the moment. This kind of behaviour is repeated throughout the whole training.
> > >
> > > **Q9: Preoscillatory regime.**
> > >
> > > ***A9:*** Thank you for noting a typo! In our experience, gradient descent does not start to oscillate upon crossing $1/\lambda_{\max}$, and indeed the dynamics on the local quadratic Taylor approximation are still contractive. Therefore, we believe that the preoscillatory regime is not present with Euclidean GD.
> > >
> > > **Q10: Evolution of eigenvalues for non-Euclidean GD.**
> > >
> > > ***A10:*** In light of our A2, it isn't clear whether there is some natural generalization of "second eigenvalue" for the non-Euclidean case. For example, for sign descent, it is easy to construct certain quadratics for which the update has only a single invariant direction. So there is a "first eigenvalue", but if there are no other invariant directions, then there is no second eigenvalue! You may say that we could just look at the Euclidean eigenvalues even for non-Euclidean GD, but we found empirically that these values do not precisely correspond with non-monotonicity in the loss. Another possibility is just to look at the "non-Euclidean eigenvalues" corresponding to whatever invariant directions do exist. The problem with this approach is that we do not know how to compute any invariant directions (or their corresponding eigenvalues) besides the maximizer of the Rayleigh quotient. So at present, we do not see a natural way to measure or even define corresponding quantities in the non-Euclidean case. You mentioned that you might have an idea, if so, we are very happy to discuss it. Thanks again.

---

> ### Comment · Reviewer_QE9g · 2025-11-25
> **I ran experiments and do not (always) match your claims**
>
> Thanks a lot for responding, to the authors.
>
> I'll comment on your rebuttals later, just wanted to comment another thing:
>
> I tried to run some cifar10 experiments, to realize that actually your results do not hold for the step sizes of the order of magnitude used by, e.g., Cohen et al (2021). Your results do hold for extremely small step sizes only in my experiments.
>
> Precisely, signGD with stepsize $10^{-4}$ or $10^{-5}$ works as you suggest, but with step size $10^{-2}$ it does not, your generalized sharpness is at 400 after 600 steps and keeps going up. Happy to share my plots if there is a way to.
>
> Can you please comment on this?
> You should add new plots in the appendix also for $10^{-1}$ step size and for wider ranges of them.

---

> ### Author Response · Authors · 2025-11-26
> **Response**
>
> We would like to clarify that $\ell_{\infty}$ descent has the following update rule
> $\mathbf{w}_{t+1} = \mathbf{w}_t - \eta\\|\nabla \mathcal{L}(\mathbf{w}_t)\\|_1\mathrm{sign}(\nabla \mathcal{L}(\mathbf{w}_t))$. In our experience, the gradient $\ell_1$-norm is significant and requires smaller values of $\eta$ when we use it for SignGD (without the gradient $\ell_1$-norm). Therefore, using values of $\eta$ as the reviewer suggests (e.g., $10^{-1}$) would lead to divergence.
>
> Please let us know if there are further questions.

---

### Official Review · Reviewer_XK99 · 2025-11-01

**Soundness:** 3
**Presentation:** 3
**Contribution:** 2
**Rating:** 6
**Confidence:** 3

**Summary:**

This paper studies the Edge of Stability (EoS) phenomenon, where sharpness hovers near $2/\eta$, and generalizes it from standard Gradient Descent (GD) to a broad class of non-Euclidean steepest-descent methods defined with respect to different norms (e.g., $\ell_\infty$-descent/SignGD, Block Coordinate Descent, Spectral GD/Muon). Using the notion of directional smoothness, the authors show that if the loss decreases and gradient norms remain roughly stable, this quantity (and the corresponding generalized sharpness) increases toward $2/\eta$ and then oscillates around it, extending the EoS picture beyond the $\ell_2$ setting. The proposed generalized sharpness $S_{\|\cdot\|}$ recovers the Euclidean and preconditioned cases as special instances. However, computing it exactly is generally intractable, and a Frank–Wolfe routine is used to estimate it. The authors further prove convergence for $\eta<2/S_{\|\cdot\|}$ and demonstrate divergence for $\eta>2/S_{\|\cdot\|}$ on simplified quadratic models. Empirically, across MLPs, CNNs, and Transformers under multiple geometries, $S_{\|\cdot\|}$ indeed tracks the $2/\eta$ threshold.

**Strengths:**

-	Understanding the training dynamics of algorithms for deep learning, for example in EOS regime, is an interesting research question.
-	The paper provides a unified analysis of multiple optimization algorithms under different norms and geometries for the Edge of Stability (EoS) phenomenon, which is interesting and appears to be novel.
-	The paper is overall well-written and easy to follow.

**Weaknesses:**

-	The theoretical results in Section 5 are derived only for a simplified quadratic model, and the results for general norms are weaker than those for the $\ell_2$ norm.
-	Estimating the generalized sharpness is computationally expensive and requires approximation.

**Questions:**

-	The discussion at the bottom of page 3 does not seem to fully explain the progressive sharpening observed in the EoS regime and is somewhat confusing. The argument presented appears to apply only to the case where the loss no longer decreases (e.g., near a minimum). However, progressive sharpening occurs well before convergence, as also shown in the paper’s plots. It would be helpful to provide a clearer explanation of this behavior.
-	In Section 3, is the reported sharpness computed using the full Hessian or a layer-wise Hessian approximation?
-	Several plots (e.g., Transformer experiment in Figures 2 and 4, and CNN experiment in Figure 5) appear more unstable and deviate from the $2/\eta$ threshold compared with other methods. Could the authors provide an intuitive explanation for this difference?
-	The experimental setups seem inconsistent across methods: for example, some CNN experiments use CIFAR-10 while Transformer experiments use Tiny Shakespeare. What motivates these choices?

---

> ### Author Response · Authors · 2025-11-21
> **Rebuttals (part 1)**
>
> We thank the reviewer for the thoughtful and encouraging comments. We are grateful for the recognition of the novelty of our study and the clarity of the paper.
>
> We hope that the explanations provided below will assist in re-evaluating our submission and possibly revising the score.
>
> **Q1: Theoretical results only for a quadratic model.**
>
> ***A1:*** We acknowledge that extending the theoretical results to a broader class of functions would further strengthen our work. However, analyzing the convergence/divergence behavior of non-Euclidean descent methods is already a challenging problem even in the quadratic case, and it is an interesting research direction in its own right. We also recognize that our understanding of non-Euclidean descent methods on quadratics remains incomplete, as their analysis is significantly more difficult than that of Euclidean descent, even for quadratic objectives, since the spectral quantities of the Hessian in this case are less studied (e.g., What are the generalizations of eigenvectors? Do they form an orthogonal basis?, etc.).
>
> Nevertheless, our theoretical results on quadratics align well with our empirical findings, particularly in supporting our choice of the generalized sharpness measure. Moreover, we emphasize that quadratic functions serve as a standard and insightful model for studying the EoS phenomenon, since any smooth function is locally quadratic. They provide an intuitive framework for understanding EoS behavior (see Figures 6 and H.4). More complex models considered in the literature (e.g., Cohen et al., 2024, Understanding Optimization in Deep Learning with Central Flow) are primarily used to provide more insights about EoS phenomenon rather than formal theoretical guarantees.
>
> **Q2: Generalized sharpness estimation is expensive.**
>
> ***A2:*** This is an important point for discussion. Indeed, computing the generalized sharpness is a non-trivial task and may even be NP-hard for certain problems. In such cases, approximation methods are required. We propose a simple procedure for estimating the generalized sharpness by running the Frank–Wolfe algorithm and computing multiple Hessian–vector products, which can be implemented efficiently in practice. The method provides reliable estimates when appropriate hyperparameters are chosen. In our view, this scheme is sufficiently robust for a first study exploring the EoS of non-Euclidean descent algorithms. Further improvements in the estimation procedure would require more advanced theoretical and practical developments, which we leave for future work (e.g., leveraging modern parallelization frameworks and incorporating layer-wise network structures).
>
> Nonetheless, such an increase in computational cost is to be expected, as even running non-Euclidean gradient descent typically requires more resources than standard Euclidean gradient descent (e.g., due to the need to approximate the polar factor in Spectral Descent). However, this additional cost often comes with a substantial improvement in performance (as observed, for instance, with Muon). These practical gains motivate the exploration of more advanced methods for estimating the step in non-Euclidean gradient descent. We believe that a similar rationale applies to the estimation of generalized sharpness.
>
> **Q3: Discussion on directional smoothness.**
>
> ***A3:*** We agree that this discussion could be clearer. Assuming that the gradient norm is strictly larger than $0$, which is the case when training neural networks, the equivalence
> $$\\mathcal{L}(\mathbf{w}\_{t+1}) \le \mathcal{L}(\mathbf{w}\_t)
> \quad\Longleftrightarrow\quad
> D(\mathbf{w}\_t,\mathbf{w}\_{t+1}) \le \frac{2}{\eta}$$
> tells us, is that if the loss is decreasing, we know that directional smoothness starts below $2/\eta$. If the loss increases, then directional smoothness must be above $2/\eta.$ Thus, if the loss initially decreases and then starts to oscillate, as is often observed in training, then directional smoothness must start below $2/\eta$ and then increase (sharpen) up to $2/\eta$, and then oscillate around $2/\eta$. We rewrote this explanation in the revision (highlighted in blue). Furthermore, we would like to be transparent that this does not explain EoS; instead, it translates the EoS phenomena to what must happen with the loss function.
>
> **Q4: Do experiments use the full Hessian?**
>
> ***A4:*** In all our experiments, we use the full Hessian in the estimation of the generalized sharpness. However, exploring a layer-wise or Fisher approximations of the full Hessian is an interesting direction for future work to make the estimation scheme cheaper to run in practice. We thank the reviewer for this valuable comment.

---

> > ### Author Response · Authors · 2025-11-21
> > **Rebuttals (part 2)**
> >
> > **Q5: Deviation from $2/\eta$ threshold.**
> >
> > ***A5:*** We thank the reviewer for this valuable comment. It is, in fact, a standard phenomenon that also happens in the Euclidean case. We explain this phenomenon in detail in Appendix B. Shortly, such a gap between the generalized sharpness and the stability threshold $2/\eta$ happens because of several unstable directions along which the oscillations are happening. Taking the max operation over several processes oscillating around the stability threshold $2/\eta$ results in a quantity, which might be higher than $2/\eta$.
> >
> > **Q6: Dataset choice motivation.**
> >
> > ***A6:*** Let us clarify the experimental setup in more detail. Our experiments largely follow the setup of Cohen et al. (2021). For the MLP and CNN models, we use the CIFAR-10 dataset (or a subset of 5,000 samples, denoted as CIFAR10-5k). For the MLP model, the images are flattened into vectors. In the case of the Transformer model, text data is required; therefore, we employ the Tiny Shakespeare dataset, which is a widely used benchmark for language modeling with small-scale models. Finally, we note that training MLP and CNN models on the CIFAR10-5k subset is too easy for Spectral Descent. Consequently, we report results on the full CIFAR-10 dataset for these cases.

---

### Meta-Review · Area_Chair_X67w · 2026-01-07

**Summary:**

The submission studies Edge of Stability (EoS) beyond Euclidean GD by framing many optimizers as non-Euclidean steepest descent (Defn 1.1) and its normalized variant (Def 1.2).   It uses directional smoothness (chosen as the tight “secant curvature” quantity) and shows an identity implying that when the gradient norm stays nonzero, loss decrease is equivalent to directional smoothness staying below before sharpening to  2/\eta and then oscillate around it once the loss oscillates. To connect to a spectral diagnostic, it defines the generalized sharpness which which recovers classical sharpness for l2 GD and the preconditioned-Hessian notion for Mahalanobis norms, and proposes a Frank–Wolfe heuristic to estimate it when closed forms are unavailable.

**Reviewer Concerns:**

Rev XK99

(1)  Theory is only for simplified quadratics; results for general norms are weaker than for l2.​
-- Authors acknowledge this limitation and argue quadratics are a standard/local model and that non-Euclidean dynamics are already challenging even on quadratics, but do not extend the theory beyond that scope.

(2) Generalized sharpness estimation is expensive / intractable and requires approximation.
-- Authors acknowledge possible NP-hardness and justify a Frank–Wolfe + HVP scheme as “robust enough” for a first study; no fundamental tractability improvement is provided.

(3) The “progressive sharpening” explanation (bottom of p3) is confusing and seems to apply only when loss stops decreasing; but sharpening happens earlier.
-- Authors rewrite the explanation as an implication from the loss oscillating and explicitly state this does not explain EoS causally.

(4) Minor clarity concerns and questions about experimental setup -- Addressed.


Reviewer QE9g

(1) The proposed quantities may be “certificates” not “causes”; if one believes self-stabilization/central-flow stories, wants a curvature quantity independent of direction/alignment, not a directional Rayleigh-quotient-type object.
--  Authors agree “directional smoothness is not the appropriate property in the strict sense,” and claim their generalized smoothness (Eq. 10) is independent of gradient/update direction; but they also concede that their generalized sharpness (max Rayleigh quotient) still doesn’t meet the reviewer’s “central-flow-only” interpretation of the classical edge of stability bejavior.

Rev

(2) Wants deeper discussion re Andreyev & Beneventano (2024), stochastic setting, and what the “right” invariant directions/eigen-objects are in non-Euclidean dynamics.
-- Authors give a detailed explanation that dynamics align with “invariant directions” (a non-Euclidean eigenvector analogue), but admit this theory is incomplete and beyond scope; they explicitly defer stochastic extensions.

(3) More experiments.
-- Authors argue they already cover CNN/MLP/Transformer, multiple norms, CIFAR-10 + Tiny Shakespeare, and add ablations + RMSprop experiments; but they also state (i) full-HVP cost limits scaling and (ii) Euclidean trace/eigenspectrum may be the wrong object in non-Euclidean geometry.

(4) Bridging Adam AEoS (Cohen et al., 2022) vs SignGD behavior.
-- Authors argue AEoS for Adam/RMSprop relies on an approximation that breaks as \beta_2 goes to 0, and add RMSprop experiments in the revision.

(5) Requests intuition for sudden drops (Block GD), pre-oscillatory regime, “second eigenvalue” analogue.
-- Authors provide explanations in the rebuttal, not new theory. Authors give mechanism-based explanations (block switching for drops; claim Euclidean GD lacks pre-osc regime; and explain why “second eigenvalue” is unclear in non-Euclidean case).

(6). Reviewer reports they can’t reproduce the claimed behavior at step sizes used in Cohen et al. (2021); claims it only holds for very small step sizes and asks for wider-range plots.
-- Authors basically agree that claimed behavior does require smaller \eta, they don’t add the requested new plots over wider step sizes in the thread.

Reviewer ybaT:

(1) Similar concern about experiments as by QE9g.
-- authors argue they cover CNN/MLP/Transformer, multiple non-Euclidean methods, CIFAR-10 + Tiny Shakespeare, and include ablations + RMSprop; later they add ResNet20/VGG11 CIFAR-10 results

(2) Theory is too shallow (only convex quadratics; lacks generalization to non-convex deep learning). Addressed? Only partially (mostly “scope”).
-- authors argue that directional smoothness identities apply to more general functions, but rigorous deep-learning EoS theory is lacking even in Euclidean case and non-convex theory is broadly open; they keep the convergence/divergence analysis at quadratics.

(3) Insufficient depth/mechanism; doesn’t explain why non-Euclidean EoS occurs or differences among norms.
-- Authors provide a “loss oscillations ⇒ directional smoothness oscillations” implication and add empirical evidence on norm differences + estimator sensitivity; they still admit the “exact reason for oscillations” is open

**Reviewer Scores:**

Rev XK99: the two core weaknesses they flagged—quadratic-only theory and costly/intractable sharpness estimation—are acknowledged rather than fundamentally resolved. Unlikely to increase the score.

Reviewer QE9g: unlikely to increase score.

For ybaT, additional experiments may slightly increase the score. The paper remains borderline. While the paper has an ambitious unifying diagnostic/generalization, but the key quantity is often computationally hard and the theory gap (generic divergence / mechanism) is not developed enough yet to warrant acceptance.

---

### Decision · Program_Chairs · 2026-01-26

Reject